# Unifying Likelihood-free Inference with Black-box Optimization and Beyond

**Dinghuai Zhang**[1,2] , **Jie Fu**[1,2*], **Yoshua Bengio**[1,2,3], **Aaron Courville**[1,2,3]
[1]Mila, [2]University of Montreal, [3]CIFAR Fellow
Montreal, Canada
{dinghuai.zhang, fujie}@mila.quebec

## Abstract

Black-box optimization formulations for biological sequence design have drawn recent attention due to their promising potential impact on the pharmaceutical industry. In this work, we propose to unify two seemingly distinct worlds: likelihood-free inference and black-box optimization, under one probabilistic framework. In tandem, we provide a recipe for constructing various sequence design methods based on this framework. We show how previous optimization approaches can be "reinvented" in our framework, and further propose new probabilistic black-box optimization algorithms. Extensive experiments on sequence design application illustrate the benefits of the proposed methodology.

## 1 Introduction

Discovering new drugs to fulfill specific criteria, such as binding affinity towards a given molecular target, is a fundamental problem in chemistry and the pharmaceutical industry (Hughes et al., 2011). In this work, we focus on an important subdomain: *de novo* biological sequence design. This task is challenging for two reasons: (1) the exploration space for sequences is combinatorially large; and (2) sequence usefulness is evaluated via a complicated process which usually involves time-consuming and expensive wet-lab experiments.

Despite the difficulty of this task, many approaches have been developed over the past few decades thanks to recent advances in biochemistry and machine learning. The Nobel Prize wining paradigm, *directed evolution* (Chen & Arnold, 1991), which conducts local evolutionary search under human guidance, is one of the popular techniques. Unfortunately, it is limited by its sample inefficiency and reliance on strong prior knowledge, *e.g.*, about where to mutate (Ahn et al., 2020). Furthermore, to compete with other machine learning methods (Gottipati et al., 2020), guided evolution (Yoshikawa et al., 2018; Jensen, 2019; Nigam et al., 2019) heavily relies on human intuition for designing domain-specific evolutionary operators, which may not always apply to tasks at hand.

In this work, we deem sequence design to be a black-box optimization problem, tasked with maximizing an unknown oracle function. We assume that oracle queries are limited due to the constraint on resources, such as the budgets for evaluating queries in a wet-lab. Thus, sample efficiency is crucial. We develop a probabilistic framework by reformulating the aforementioned black-box optimization target as a posterior modeling problem. With this framework, we draw a surprising connection between likelihood-free inference and sequence design, and thus linking two fields which are previously considered as unrelated. The key observation we leverage here for establishing this connection is that both settings share similar elements and targets which will be elaborated in Section 2.2. This connection facilitates our understanding of both fields and provides a recipe for developing sequence design algorithms. Going beyond, we also combine different probabilistic modeling insights and develop three novel *composite* probabilistic algorithms. We point out that our framework could actually be applied to any black-box optimization settings, but in this work we focus on its application to biological sequence design.

To demonstrate the empirical effectiveness of our methods, we conduct systematical experiments to evaluate their performance on four *in-silico* sequence design benchmarks. Our proposed meth-

---

*Corresponding Author

ods achieve at least comparable results to existing baselines, and the proposed composite methods behave consistently better than all other ones across various sequence design tasks.

We summarize our contribution as follows:

- We develop a probabilistic framework that unifies likelihood-free inference and black-box optimization.

- Based on this framework, we provide a recipe for designing algorithms for black-box problems. We apply these ideas to propose a series of composite design algorithms.

- We perform systematical evaluation on a series of black-box sequence design benchmarks, and find that these algorithms achieves consistently comparable or better results compared to previous ones, thus illustrating the benefit of the proposed unified framework.

## 2 A UNIFYING PROBABILISTIC FRAMEWORK

### 2.1 BACKGROUND

**Likelihood-free inference (LFI)**. We use $\boldsymbol{\theta} \in \Theta$ and $\mathbf{x} \in \mathcal{X}$ to separately denote the parameters and the data generated via the mechanism $\mathbf{x} \sim p(\mathbf{x}|\boldsymbol{\theta})$. In this scenario, LFI refers to a special kind of Bayesian inference setting where the likelihood function is not tractable but sampling (by simulation) from the likelihood is feasible. Consider the objective of modeling the Bayesian posterior when we cannot compute the likelihood $p(\mathbf{x}_o|\boldsymbol{\theta})$:

$$p(\boldsymbol{\theta}|\mathbf{x}_o) \propto p(\boldsymbol{\theta}) \underbrace{p(\mathbf{x}_o|\boldsymbol{\theta})}_{?}, \qquad (1)$$

where $\mathbf{x}_o$ is the observed data, $p(\boldsymbol{\theta})$ is the (given) prior over the model parameters $\boldsymbol{\theta}$, $p(\mathbf{x}|\boldsymbol{\theta})$ is the intractable likelihood function and $p(\boldsymbol{\theta}|\mathbf{x})$ is the desired posterior over $\boldsymbol{\theta}$. While we do not have access to the exact likelihood, we can still simulate (sample) data $\mathbf{x}$ from the model simulator: $\mathbf{x} \sim p(\mathbf{x}|\boldsymbol{\theta})$. Instead of trying to obtain a numerical value of the generic posterior $p(\boldsymbol{\theta}|\mathbf{x})$ for arbitrary $\mathbf{x}$, LFI only tries to obtain an approximation of $p(\boldsymbol{\theta}|\mathbf{x}_o)$ for the given $\mathbf{x}_o$. During the inference process, we can take advantage of the sampled data: $\mathcal{D} = \{(\boldsymbol{\theta}_i, \mathbf{x}_i)\}_{i=1}^n$ where $\mathbf{x}_i \sim p(\mathbf{x}|\boldsymbol{\theta}_i)$ for selected values of $\boldsymbol{\theta}_i$.

**Biological black-box sequence design**. We consider biological sequence design as a black-box optimization problem:

$$\mathbf{m}^* = \arg\max_{\mathbf{m} \in \mathcal{M}} f(\mathbf{m}),$$

where $f(\cdot)$ is the oracle score function, and we would like to discover values of $\mathbf{m}$ for which $f(\mathbf{m})$ is large. In real-world situations, a query of this oracle $f$ could represent a series of wet-lab experiments to measure specific chemical properties or specificity for a given binding site target. In general, these experiments are time- and cost-consuming. As a result, the total number of queries is limited.

In our setting, we use $\mathcal{M} = \mathcal{V}^L$ to denote the search space for sequences with fixed length $L$, where $\mathcal{V}$ is the vocabulary for each entry of the sequence: for DNA nucleotides $|\mathcal{V}| = 4$, and for protein amino acids $|\mathcal{V}| = 20$. For variable length setting, we have $\mathcal{M} = \cup_{L \in [L_{\min}, L_{\max}]} \mathcal{V}^L$, where $L_{\min}$ and $L_{\max}$ are the minimal and maximal length, respectively.

### 2.2 CONNECTING LFI AND BLACK-BOX OPTIMIZATION

In order to draw a connection to LFI, we require a probabilistic formulation of the black-box sequence design problem. To this end, we relax the goal of searching for a single maximum of the oracle / score function $f$ to a posterior modeling problem, *i.e.*, finding a representative sample of the configurations of $\mathbf{m}$ sampled with probability related to some target posterior. Think of $\mathcal{C}$ is the set of sequences with these desirable configurations, $\mathcal{E}$ is a Boolean event about whether a sequence $\mathbf{m}$ belongs to $\mathcal{C}$, and our goal is to characterize the posterior distribution $p(\mathbf{m}|\mathcal{E})$ from which we obtain the desired sequences. Below, we consider two specific ways of doing this:

**Example A.** We explicitly define $\mathcal{C}$ (and $\mathcal{E}$ accordingly) as all the sequences whose scores are larger than a given threshold $s$:

$$\mathcal{C} = \{\mathbf{m}|f(\mathbf{m}) \geq s\}. \tag{2}$$

Here $s$ could be any fixed value, or a certain quantile of a particular score distribution. In this way, we have $p(\mathcal{E}|\mathbf{m}) = p(\mathbf{m} \in \mathcal{C}|\mathbf{m}) = \mathbb{1}\{f(\mathbf{m}) \geq s\}$ where $\mathbb{1}\{\}$ is the indicator function.

**Example B.** In a softer version of $\mathcal{E}$ and $\mathcal{C}$, we can define its conditional probability of being true to follow a Boltzmann distribution:

$$p(\mathcal{E}|\mathbf{m}) = p(\mathbf{m} \in \mathcal{C}|\mathbf{m}) \propto \exp(f(\mathbf{m})/\tau). \tag{3}$$

where $\tau$ is a temperature parameter. We introduce the exponential because $f(\cdot)$ does not necessarily take positive values. Any monotone transformation of $f(\cdot)$ to non-negative reals could be used, so that sequences with larger oracle scores have a greater probability of making $\mathcal{E}$ true.

With this posterior objective, our goal now becomes effectively modeling and sampling from the posterior $p(\mathbf{m}|\mathcal{E})$. It is thus natural to resort to the tools of Bayesian inference for this task. In order to examine this possibility, we draw a detailed comparison between the settings of black-box sequence design problem and likelihood-free Bayesian inference in Table 1.

|  | Likelihood-free inference | Black-box optimization |
|---|---|---|
| Element | $(\boldsymbol{\theta}, \mathbf{x})$ | $(\mathbf{m}, s)$ |
| Target | $p(\boldsymbol{\theta}|\mathbf{x}_o)$ | $p(\mathbf{m}|\mathcal{E})$ |
| Constraint | limited simulation: $\mathbf{x} \sim p(\mathbf{x}|\boldsymbol{\theta})$
intractable likelihood: $p(\mathbf{x}|\boldsymbol{\theta})$ | limited query: $s \sim f(\mathbf{m})$
black-box oracle: $f(\mathbf{m})$ |

Table 1: Correspondence between likelihood-free inference and black-box optimization.

It can be observed that both tasks share similar elements and targets. The two settings also share similar limitations on the allowed queries, which are too time-consuming and / or cost-intensive. Notice that in sequence design, the oracle could be either exact or noisy, thus we use the more general $s \sim f(\mathbf{m})$ formulation rather than $s = f(\mathbf{m})$. We will further present several concrete examples as demonstrations of this correspondence in the following section.

Another way to understand this correspondence is to consider the following mapping $T$:

$$T : \Theta \times \mathcal{X} \to \mathcal{M} \times \mathbb{R}$$
$$(\boldsymbol{\theta}, \mathbf{x}) \mapsto (\mathbf{m}, s), \quad \text{s.t. } s = -\|\mathbf{x} - \mathbf{x}_o\|.$$

Here we can see the score value $s$ as a quantitative metric for how close the generated data $\mathbf{x}$ (given $\boldsymbol{\theta}$) is to the target observed data $\mathbf{x}_o$. In addition, querying the oracle in the sequence design setting can also be thought of as follows: (1) sample $\mathbf{x} \sim p(\cdot|\boldsymbol{\theta})$ and then (2) calculate $s = -\|\mathbf{x} - \mathbf{x}_o\|$ under some distance $\|\cdot\|$. In this manner, $T$ could conceptually transform any LFI problem into a black-box optimization task. In this work, we only focus on the application of sequence design.

## 3 METHODOLOGY

We provide a recipe for designing new sequence design algorithms based on the correspondence in Section 2.2. The recipe induces different approaches by modeling different probabilistic component of the Bayesian inference problem. We begin with common algorithm restrictions under this setting.

**Common constraint for algorithms**. Due to the restriction of simulation / query in our setting, we constrain our algorithms to act in a sequential / iterative way, gradually achieving the desired posterior round by round. Every algorithm starts with an empty dataset $\mathcal{D} = \varnothing$ and an initial proposal $p_1(\cdot) = p(\cdot)$, where $p(\cdot)$ is the prior given by the task. In the $r$-th round of this multi-round setting, the algorithm would use the proposal $p_r(\cdot)$ of this round to sample a batch of data $(\boldsymbol{\theta} / \mathbf{m})$ for simulation / query, and augment the current dataset $\mathcal{D}$ with the newly obtained batch of data. We use $n$ to denote the batch size for each round's simulation / query. Afterwards, the algorithm updates

the proposal to $p_{r+1}(\cdot)$. The outcomes for the two settings we discuss may be slightly different: an algorithm for likelihood-free inference would return the posterior, while a sequence design method would return the dataset of all the sequences it has queried, which hopefully contains desired high scored sequences. On the other hand, a sequence design method could produce as an intermediate result a generative model for sampling queries, which then completely fits with the LFI framework.

## 3.1 BACKWARD MODELING OF THE MECHANISM

Approximate Bayesian Computation (ABC) (Beaumont et al., 2002) is a standard method for tackling LFI problems. In Algorithm 1, we display one of the most popular variants: Sequential Monte Carlo-Approximate Bayesian Computation (SMC-ABC) (Beaumont et al., 2009). In each round, parameters $\boldsymbol{\theta}$ are sampled from the current proposal distribution $p_r(\boldsymbol{\theta})$ for simulation. A rejection step is then involved to remove the $\boldsymbol{\theta}_i$ whose simulation outcomes $\mathbf{x}_i$ cannot reproduce the observed data $\mathbf{x}_o$ with sufficient accuracy. The remaining accepted $\{\boldsymbol{\theta}_i\}_i$ are adopted to update the next round's proposal $p_{r+1}(\cdot)$ towards the target posterior, *i.e.*, by refitting $q_\phi$ with the modified data. We defer more details of this approach to Section A.1 in Appendix.

---

**Algorithm 1** SMC-ABC

$p_1(\boldsymbol{\theta}) \leftarrow p(\boldsymbol{\theta})$;
**for** $r$ in 1 to $R$ **do**
   **repeat**
      sample $\boldsymbol{\theta}_i \sim p_r(\boldsymbol{\theta})$;
      simulate $\mathbf{x}_i \sim p(\mathbf{x}|\boldsymbol{\theta}_i)$;
   **until** $n$ samples are obtained
   $\mathcal{D} \leftarrow \mathcal{D} \cup \{(\boldsymbol{\theta}_i, \mathbf{x}_i)\}_{i=1}^n$
   sort $\mathcal{D}$ according to $-\|\mathbf{x}_i - \mathbf{x}_o\|$;
   fit $q_\phi(\boldsymbol{\theta})$ with top $\{\boldsymbol{\theta}_i\}_i$ in $\mathcal{D}$;
   $p_{r+1}(\boldsymbol{\theta}) \leftarrow q_\phi(\boldsymbol{\theta})$;
**end for**
**return** $\hat{p}(\boldsymbol{\theta}|\mathbf{x}_o) = p_{R+1}(\boldsymbol{\theta})$

---

**Algorithm 2** FB-VAE

$p_1(\mathbf{m}) \leftarrow p(\mathbf{m})$;
**for** $r$ in 1 to $R$ **do**
   **repeat**
      sample $\mathbf{m}_i \sim p_r(\mathbf{m})$;
      query the oracle: $s_i \leftarrow f(\mathbf{m}_i)$;
   **until** $n$ samples are obtained
   $\mathcal{D} \leftarrow \mathcal{D} \cup \{(\mathbf{m}_i, s_i)\}_{i=1}^n$;
   sort $\mathcal{D}$ according to $s_i$
   fit $q_\phi(\mathbf{m})$ with top $\{\mathbf{m}_i\}_i$ in $\mathcal{D}$;
   $p_{r+1}(\mathbf{m}) \leftarrow q_\phi(\mathbf{m})$;
**end for**
**return** $\{\mathbf{m} : (\mathbf{m}, s) \in \mathcal{D}\}$

---

It would then be natural to construct an analogical sequence design algorithm using top scored entities $\{\mathbf{m}_i\}_i$ to guide the update of a certain sequence distribution, see Algorithm 2. Interestingly, this is the proposed sequence design algorithm in Gupta & Zou (2019), where the authors name this kind of updating "feedback" because training of the parametric generator $q_\phi(\mathbf{m})$ exploits feedback signals from the oracle. In this paper, we follow Brookes & Listgarten (2018) to crystallize $q_\phi(\mathbf{m})$ to be a variational autoencoder (Kingma & Welling, 2014), and use the term Feedback-Variational AutoEncoder (FB-VAE) to refer to Algorithm 2. We place Algorithm 1 & 2 side-by-side to highlight their correspondence. We also make the same arrangement for the following Algorithm 3 & 4, Algorithm 5 & 6 and Algorithm 7 & 8.

---

**Algorithm 3** Sequential Neural Posterior

$p_1(\boldsymbol{\theta}) \leftarrow p(\boldsymbol{\theta})$;
**for** $r$ in 1 to $R$ **do**
   **repeat**
      sample $\boldsymbol{\theta}_i \sim p_r(\boldsymbol{\theta})$;
      simulate $\mathbf{x}_i \sim p(\mathbf{x}|\boldsymbol{\theta}_i)$;
   **until** $n$ samples are obtained
   $\mathcal{D} \leftarrow \mathcal{D} \cup \{(\boldsymbol{\theta}_i, \mathbf{x}_i)\}_{i=1}^n$;
   $q_\phi \leftarrow \arg\min_q \mathbb{E}_{\mathbf{x}}\left[D_{\mathrm{KL}}(p(\boldsymbol{\theta}|\mathbf{x})\|q)\right]$;
   $p_{r+1}(\boldsymbol{\theta}) \leftarrow q_\phi(\boldsymbol{\theta}|\mathbf{x}_o)$;
**end for**
**return** $\hat{p}(\boldsymbol{\theta}|\mathbf{x}_o) = p_{R+1}(\boldsymbol{\theta})$

---

**Algorithm 4** Design by Adaptive Sampling

$p_1(\mathbf{m}) \leftarrow p(\mathbf{m})$;
**for** $r$ in 1 to $R$ **do**
   **repeat**
      sample $\mathbf{m}_i \sim p_r(\mathbf{m})$;
      query the oracle: $s_i \leftarrow f(\mathbf{m}_i)$;
   **until** $n$ samples are obtained
   $\mathcal{D} \leftarrow \mathcal{D} \cup \{(\mathbf{m}_i, s_i)\}_{i=1}^n$;
   $q_\phi \leftarrow \arg\min_q D_{\mathrm{KL}}(p(\mathbf{m}|\mathcal{E})\|q)$;
   $p_{r+1}(\mathbf{m}) \leftarrow q_\phi(\mathbf{m})$;
**end for**
**return** $\{\mathbf{m} : (\mathbf{m}, s) \in \mathcal{D}\}$

---

In comparison with SMC-ABC, the Sequential Neural Posterior (SNP) method (Papamakarios & Murray, 2016; Lueckmann et al., 2017; Greenberg et al., 2019) for likelihood-free inference adopts a more flexible approach, taking the power of conditional neural density estimator (*e.g.*, Papamakarios et al. (2017)) to model the general posterior $p(\boldsymbol{\theta}|\mathbf{x})$, which takes arbitrary $\boldsymbol{\theta}$ and $\mathbf{x}$ as two inputs and outputs a distribution. This neural estimator is trained via approximately minimizing the

Kullback–Leibler (KL) divergence between $q_\phi(\boldsymbol{\theta}|\mathbf{x})$ and the true posterior $p(\boldsymbol{\theta}|\mathbf{x})$. We defer more training details to Section A.1 in Appendix. Under the connection viewpoint, one similar algorithm for sequence design is the Design by Adaptive Sampling (DbAS) proposed in Brookes & Listgarten (2018) which is characterized in Algorithm 4, fitting $q_\phi(\mathbf{m})$ through minimizing the KL divergence with the posterior $p(\mathbf{m}|\mathcal{E})$. Based on the difference in specific implementations, both algorithms have more than one variant, whose details are deferred to Section A.1 in Appendix.

We refer to the above algorithms as "backward modeling" because the trained generative network $q_\phi$ (going from $\mathbf{x}/\mathcal{E}$ to $\boldsymbol{\theta}/\mathbf{m}$) is a sort of reverse model of the simulation mechanism (which goes from $\boldsymbol{\theta}/\mathbf{m}$ to $\mathbf{x}/s$).

## 3.2 Forward modeling of the mechanism

Whereas the above methods focus on directly modeling the target posterior with a generative model that learns a "reverse mechanism" of the simulation process, it is also possible to model the "forward mechanism", which is consistent with the simulation process. Papamakarios et al. (2019) claim that the forward modeling approach may be an easier task than its backward counterpart, as unbiased estimation of the likelihood does not depend on the choice of proposal. Consequently, in contrast to SNP, Papamakarios et al. (2019) chooses to train a neural density estimator to model the conditional likelihood distribution $q_\phi(\mathbf{x}|\boldsymbol{\theta})$ sequentially in each round. The training is achieved by maximizing the total log likelihood $\max_q \sum_i \log q_\phi(\mathbf{x}_i|\boldsymbol{\theta}_i)$ with data samples from the dataset $\mathcal{D}$ at the current ($r$-th) round. The downside of this forward approach is an additional computational Markov Chain Monte Carlo (MCMC) step is needed to sample from the $r$-th round posterior / proposal $p_r(\boldsymbol{\theta})$. The resulting approach, which is coined (Papamakarios et al., 2019) the Sequential Neural Likelihood (SNL), is summarized in Algorithm 5.

In the spirit of directly modeling the forward mechanism of sequence design, we train a regressor $\hat{f}_\phi(\mathbf{m})$ in a supervised manner to fit the oracle scorer. In order to adapt this regressor into the update procedure of the proposal of the next round, we use $\tilde{q}(\mathbf{m})$ to denote the unknown posterior $p(\mathbf{m}|\mathcal{E})$ with knowledge of $\hat{f}_\phi(\mathbf{m})$ and prior $p(\mathbf{m})$. The specific construction of $\tilde{q}(\mathbf{m})$ depends on the choice of $\mathcal{E}$. For instance, if we choose Example B in Section 2.2 to be the definition of $\mathcal{E}$, then $\tilde{q}(\mathbf{m})$ is the distribution with (unnormalized) probability $p(\mathbf{m}) \cdot \exp(\hat{f}_\phi(\mathbf{m})/\tau)$. See Section A.2 in Appendix for more elaboration about this point. We then choose the update procedure of the proposal $p_{r+1}(\mathbf{m})$ to be analogical to that of SNL. We name this proposed algorithm to be Iterative Scoring (IS) to avoid confusion with likelihood-free inference algorithms. Furthermore, depending on different definition of $\mathcal{E}$, we use the name "IS-A" and "IS-B" for them in the following sections.

---

**Algorithm 5** Sequential Neural Likelihood

$p_1(\boldsymbol{\theta}) \leftarrow p(\boldsymbol{\theta})$;
**for** $r$ in 1 to $R$ **do**
   **repeat**
      sample $\boldsymbol{\theta}_i \sim p_r(\boldsymbol{\theta})$;
      simulate $\mathbf{x}_i \sim p(\mathbf{x}|\boldsymbol{\theta}_i)$;
   **until** $n$ samples are obtained
   $\mathcal{D} \leftarrow \mathcal{D} \cup \{(\boldsymbol{\theta}_i, \mathbf{x}_i)\}_{i=1}^n$
   fit $q_\phi(\mathbf{x}|\boldsymbol{\theta})$ with $\mathcal{D}$;
   $p_{r+1}(\boldsymbol{\theta}) \propto p(\boldsymbol{\theta}) \cdot q_\phi(\mathbf{x}_o|\boldsymbol{\theta})$;
**end for**
**return** $\hat{p}(\boldsymbol{\theta}|\mathbf{x}_o) = p_{R+1}(\boldsymbol{\theta})$

---

**Algorithm 6** Iterative Scoring

$p_1(\mathbf{m}) \leftarrow p(\mathbf{m})$;
**for** $r$ in 1 to $R$ **do**
   **repeat**
      sample $\mathbf{m}_i \sim p_r(\mathbf{m})$;
      query the oracle: $s_i \leftarrow f(\mathbf{m}_i)$;
   **until** $n$ samples are obtained
   $\mathcal{D} \leftarrow \mathcal{D} \cup \{(\mathbf{m}_i, s_i)\}_{i=1}^n$;
   fit $\hat{f}_\phi(\mathbf{m})$ with $\mathcal{D}$;
   construct $\tilde{q}(\mathbf{m})$ with $\hat{f}_\phi(\cdot)$ and $p(\mathbf{m})$;
   $p_{r+1}(\mathbf{m}) \leftarrow \tilde{q}(\mathbf{m})$;
**end for**
**return** $\{\mathbf{m} : (\mathbf{m}, s) \in \mathcal{D}\}$

---

## 3.3 Modeling a probability ratio

In this subsection, we discuss yet another approach, through the estimation of a probability ratio. Gutmann & Hyvärinen (2010) proposes noise contrastive estimation as a statistical inference approach. This methodology turns a hard probability modeling problem into binary classification, which is considered easier to learn. In contrast to the aforementioned likelihood-free inference methods which rely on a form of density estimation to perform the task, Sequential Neural Ratio

(SNR) (Hermans et al., 2019) takes a similar approach as noise contrastive estimation. SNR adopts a classification approach to estimate the likelihood-to-evidence ratio $r(\boldsymbol{\theta}, \mathbf{x}) = p(\boldsymbol{\theta}|\mathbf{x})/p(\boldsymbol{\theta})$. SNR is summarized in Algorithm 7. Specifically, in each round, SNR fits a binary classifier $d_\phi(\boldsymbol{\theta}, \mathbf{x}) \in [0, 1]$ in the following manner:

$$\arg\min_{d} \left\{ \sum_{(\boldsymbol{\theta}_i, \mathbf{x}_i) \in \mathcal{D}} [-\log d(\mathbf{x}_i, \boldsymbol{\theta}_i)] + \sum_{(\boldsymbol{\theta}'_i, \mathbf{x}_i) \in \mathcal{D}'} [-\log(1 - d(\mathbf{x}_i, \boldsymbol{\theta}'_i))] \right\}. \tag{4}$$

We show that with the $\mathcal{D}$ and $\mathcal{D}'$ established in Algorithm 7, we have

$$d^*(\boldsymbol{\theta}, \mathbf{x}) = \frac{p(\boldsymbol{\theta}|\mathbf{x})}{p(\boldsymbol{\theta}) + p(\boldsymbol{\theta}|\mathbf{x})}, \quad r^*(\boldsymbol{\theta}, \mathbf{x}) := \frac{d^*(\boldsymbol{\theta}, \mathbf{x})}{1 - d^*(\boldsymbol{\theta}, \mathbf{x})} = \frac{p(\boldsymbol{\theta}|\mathbf{x})}{p(\boldsymbol{\theta})}$$

where the "$*$" denotes the optimality. We have the following Proposition 1:

**Proposition 1.** *Let $p_0(\mathbf{a})$ and $p_1(\mathbf{a})$ be two distributions for $d_a$-dimension random variable $\mathbf{a}$ which takes value in the space of $\mathcal{A} = \mathbb{R}^{d_a}$, and $d(\mathbf{a}) : \mathcal{A} \to [0, 1]$ is a real-value function mapping any $\mathbf{a}$ to a positive real value number. Then the functional optimization problem*

$$\arg\max_{d:\mathcal{A}\to[0,1]} \left\{ \mathbb{E}_{\mathbf{a}\sim p_0(\mathbf{a})}[\log d(\mathbf{a})] + \mathbb{E}_{\mathbf{a}\sim p_1(\mathbf{a})}[\log(1 - d(\mathbf{a}))] \right\}.$$

*will lead to the optimal solution $d^*(\mathbf{a}) = \frac{p_0(\mathbf{a})}{p_0(\mathbf{a})+p_1(\mathbf{a})}$.*

See the proof in Section A.3 in Appendix. After training $d$, SNR can obtain the posterior density value by $p(\boldsymbol{\theta}|\mathbf{x}) = r^*(\boldsymbol{\theta}, \mathbf{x})p(\boldsymbol{\theta})$. SNR mirrors SNL in that it samples the new proposal $p_{r+1}(\boldsymbol{\theta})$ without explicitly modeling the posterior.

On the other hand, we propose a sequence design algorithm analogous to SNR and named Iterative Ratio (IR), which estimates the probability ratio $r(\mathbf{m}) = p(\mathbf{m}|\mathcal{E})/p(\mathbf{m})$ for the purpose of posterior sampling. IR first builds two datasets $\mathcal{D}$ and $\mathcal{D}'$, corresponding to two different distributions $p(\mathbf{m}|\mathcal{E})$ and $p(\mathbf{m})$. We take a similar binary classification approach, whose training objective is $\min_d \left\{ \sum_{\mathbf{m}\in\mathcal{D}}[-\log d(\mathbf{m})] + \sum_{\mathbf{m}\in\mathcal{D}'}[-\log(1 - d(\mathbf{m}))] \right\}$. The desired ratio is then obtained by $r(\mathbf{m}) := d(\mathbf{m})/(1 - d(\mathbf{m}))$. Other components of IR follow the scheme of IS, and IR is schematized in Algorithm 8. We point out that IR does not have two variants as IS does, as the definition for $\mathcal{E}$ in Example B is not usable because of the unknown normalizing constant for probability $p(\mathcal{E}|\mathbf{m})$. See Section A.3 in Appendix for more explanation.

---

**Algorithm 7** Sequential Neural Ratio

$p_1(\boldsymbol{\theta}) \leftarrow p(\boldsymbol{\theta})$;
**for** $r$ in 1 to $R$ **do**
  **repeat**
    sample $\boldsymbol{\theta}_i, \boldsymbol{\theta}'_i \sim p_r(\boldsymbol{\theta})$;
    simulate $\mathbf{x}_i \sim p(\mathbf{x}|\boldsymbol{\theta}_i)$;
  **until** $n$ samples are obtained
  $\mathcal{D} \leftarrow \mathcal{D} \cup \{(\boldsymbol{\theta}_i, \mathbf{x}_i)\}_{i=1}^n$
  $\mathcal{D}' \leftarrow \mathcal{D}' \cup \{(\boldsymbol{\theta}'_i, \mathbf{x}_i)\}_{i=1}^n$
  train $d_\phi(\boldsymbol{\theta}, \mathbf{x})$ classifying between $\mathcal{D}$ and $\mathcal{D}'$
  with the loss in Eq. 4;
  $r_\phi(\boldsymbol{\theta}, \mathbf{x}) \leftarrow \frac{d_\phi(\boldsymbol{\theta}, \mathbf{x})}{1-d_\phi(\boldsymbol{\theta}, \mathbf{x})}$;
  $p_{r+1}(\boldsymbol{\theta}) \propto r_\phi(\boldsymbol{\theta}, \mathbf{x}) \cdot p(\boldsymbol{\theta})$;
**end for**
**return** $\hat{p}(\boldsymbol{\theta}|\mathbf{x}_o) = p_{R+1}(\boldsymbol{\theta})$

**Algorithm 8** Iterative Ratio

$p_1(\mathbf{m}) \leftarrow p(\mathbf{m})$;
**for** $r$ in 1 to $R$ **do**
  **repeat**
    sample $\mathbf{m}_i \sim p_r(\mathbf{m})$;
    query the oracle: $s_i \leftarrow f(\mathbf{m}_i)$;
  **until** $n$ samples are obtained
  $\mathcal{D} \leftarrow \mathcal{D} \cup \{(\mathbf{m}_i, s_i)\}_{i=1}^n$;
  construct $\tilde{\mathcal{D}}$ with $\mathbf{m}$ in $\mathcal{D}$ satisfying $\mathcal{E}$;
  construct $\tilde{\mathcal{D}}'$ from $p(\mathbf{m})$;
  train $d_\phi(\mathbf{m})$ classifying between $\tilde{\mathcal{D}}$ and $\tilde{\mathcal{D}}'$;
  $r_\phi(\mathbf{m}) \leftarrow \frac{d_\phi(\mathbf{m})}{1-d_\phi(\mathbf{m})}$;
  $p_{r+1}(\boldsymbol{\theta}) \propto r_\phi(\mathbf{m}) \cdot p(\mathbf{m})$;
**end for**
**return** $\{\mathbf{m} : (\mathbf{m}, s) \in \mathcal{D}\}$

---

Interestingly, a recent SNR-like work, EG-LF-MCMC (Begy & Schikuta, 2021), proposes to train the classifier on tuples of $(\boldsymbol{\theta}, \epsilon = \|\mathbf{x} - \mathbf{x}_o\|)$ instead of $(\boldsymbol{\theta}, \mathbf{x})$. This algorithm can be also seen as a more precise analogy of our Iterative Ratio in the LFI context, as we have pointed out in Section 2.2 that a conceptual link could be drawn between $s$ and $-\|\mathbf{x} - \mathbf{x}_o\|$.

### 3.4 COMPOSITE PROBABILISTIC METHODS

Building on the above analogies and framework, we move beyond the above algorithms in this subsection. The previous lines of approach – the direct posterior modeling methods in Section 3.1 and the indirect methods in Section 3.2 and 3.3 – both have their own advantages and disadvantages. The former methods may fail to get accurate inference result due to unmatched proposals, while the latter ones would need extra large amount of computation for the MCMC sampling process before obtaining accurate posterior samples, etc. Here we study composite algorithms that combine the aforementioned ingredients through the lens of our proposed unified framework. Our goal is to combine the strengths from both kinds of methods.

We first introduce Iterative Posterior Scoring (IPS) method illustrated in Algorithm 9. IPS also uses a neural network $\hat{f}_\phi(\mathbf{m})$ to model the forward mechanism as IS does, and again we use $\tilde{q}$ here to denote the target posterior $p(\mathbf{m}|\mathcal{E})$. Instead of applying computational MCMC steps here, we train a second parametrized model $q_\psi$ to model $\tilde{q}$ by minimizing the KL divergence between them. Notice that the optimization of $q_\psi$ is restricted within a neural network parameterization family. As a result, in the next round, we can directly utilize $q_\psi(\mathbf{m})$ to serve as a flexible generative proposal of $p_{r+1}(\mathbf{m})$. Like IS, the IPS algorithm also has two different variants with regard to different choices of $\mathcal{E}$, we name them to be IPS-A and IPS-B. Two choices differ in the detailed construction of distribution $\tilde{q}(\mathbf{m})$.

In a similar spirit, we propose the Iterative Posterior Ratio (IPR) algorithm (see Algorithm 10). IPR is close to the IR algorithm in many aspects, but also adopts a second neural network model $q_\psi(\mathbf{m})$ like IPS. IPR works similarly to IR, in that we also construct $\tilde{q}(\mathbf{m})$, taking advantage of $r_\phi(\mathbf{m})$ and the prior $p(\mathbf{m})$ simply via $\tilde{q}(\mathbf{m}) \leftarrow r_\phi(\mathbf{m}) \cdot p(\mathbf{m})$. Note that the usage of two models in IPR is not exactly the same as in IPS: $q_\psi(\mathbf{m})$ is also achieved via minimizing KL divergence with $\tilde{q}(\mathbf{m})$, but the training of model $d_\phi(\mathbf{m})$ is closer to that in IR rather than the $\hat{f}_\phi(\mathbf{m})$ in IS. Another similarity between IPR and IR is that IPR also only has one variant, since the Example B is not applicable for this ratio modeling approach (see Section A.4 in Appendix for more details).

---

**Algorithm 9** Iterative Posterior Scoring

$p_1(\mathbf{m}) \leftarrow p(\mathbf{m})$;
**for** $r$ in 1 to $R$ **do**
  **repeat**
    sample $\mathbf{m}_i \sim p_r(\mathbf{m})$;
    query the oracle: $s_i \leftarrow f(\mathbf{m}_i)$;
  **until** $n$ samples are obtained
  $\mathcal{D} \leftarrow \mathcal{D} \cup \{(\mathbf{m}_i, s_i)\}_{i=1}^n$;
  fit $\hat{f}_\phi(\mathbf{m})$ with $\mathcal{D}$;
  construct $\tilde{q}(\mathbf{m})$ with $\hat{f}_\phi(\cdot)$ and $p(\mathbf{m})$;
  $q_\psi \leftarrow \arg\min_q D_{\mathrm{KL}}(\tilde{q}(\mathbf{m})\|q)$;
  $p_{r+1}(\mathbf{m}) \leftarrow q_\psi(\mathbf{m})$;
**end for**
**return** $\{\mathbf{m} : (\mathbf{m}, s) \in \mathcal{D}\}$

---

**Algorithm 10** Iterative Posterior Ratio

$p_1(\mathbf{m}) \leftarrow p(\mathbf{m})$;
**for** $r$ in 1 to $R$ **do**
  **repeat**
    sample $\mathbf{m}_i \sim p_r(\mathbf{m})$;
    query the oracle: $s_i \leftarrow f(\mathbf{m}_i)$;
  **until** $n$ samples are obtained
  $\mathcal{D} \leftarrow \mathcal{D} \cup \{(\mathbf{m}_i, s_i)\}_{i=1}^n$;
  construct $\tilde{\mathcal{D}}$ with $\mathbf{m}$ in $\mathcal{D}$ satisfing $\mathcal{E}$;
  construct $\tilde{\mathcal{D}}'$ from $p(\mathbf{m})$;
  train $d_\phi(\mathbf{m})$ classifying between $\tilde{\mathcal{D}}$ and $\tilde{\mathcal{D}}'$;
  $r_\phi(\mathbf{m}) \leftarrow \frac{d_\phi(\mathbf{m})}{1-d_\phi(\mathbf{m})}$;
  construct $\tilde{q}(\mathbf{m})$ with $r_\phi(\mathbf{m})$ and $p(\mathbf{m})$;
  $q_\psi \leftarrow \arg\min_q D_{\mathrm{KL}}(\tilde{q}(\mathbf{m})\|q)$;
  $p_{r+1}(\mathbf{m}) \leftarrow q_\psi(\mathbf{m})$;
**end for**
**return** $\{\mathbf{m} : (\mathbf{m}, s) \in \mathcal{D}\}$

---

## 4 EXPERIMENTS

### 4.1 SETUP

In this section, we systematically evaluate the proposed methods and baselines on four different *in-silico* biological sequence design benchmarks. In every round, we allow each algorithm to query the black-box oracle for a batch of $n$ sequences $\mathbf{m}_i$ to obtain their true scores $s_i$, with $n = 100$ for all experiments. The total number of rounds differs across different tasks.

We experiment with our six proposed methods: Iterative Scoring (-A/B) from Section 3.2, Iterative Ratio from Section 3.3, Iterative Posterior Scoring (-A/B) and Iterative Posterior Ratio from Sec-

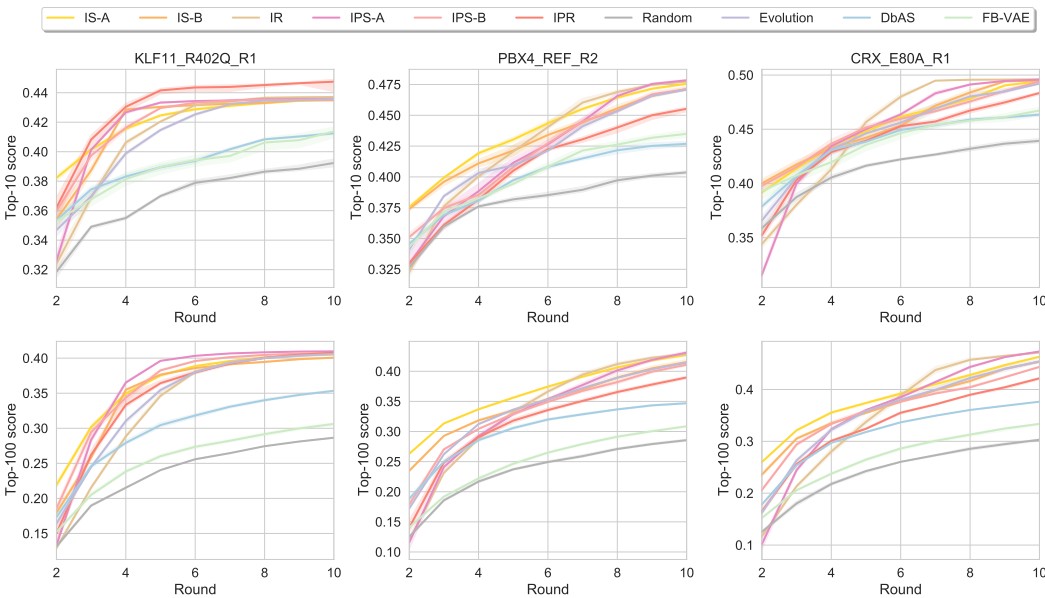

Figure 1: Top score (y-axis) curves of different methods on 3 TfBind problems (KLF11_R402Q_R1, PBX4_REF_R2 and CRX_E80A_R1) with regard to the number of rounds.

|  | IS-A | IS-B | IR | IPS-A | IPS-B | IPR | RANDOM | EVOLUTION | DBAS | FB-VAE |
|---|---|---|---|---|---|---|---|---|---|---|
| TOP-10 | 8.43 | 6.93 | 4.79 | 6.21 | 8.36 | 7.00 | 1.14 | 4.36 | 5.07 | 2.71 |
| TOP-100 | 9.57 | 7.93 | 4.07 | 6.71 | 7.64 | 6.00 | 1.00 | 5.50 | 4.57 | 2.00 |

Table 2: Mean rank of evaluated algorithms with regard to the area under the top score curve for TfBind problems. The rank ranges from 1 to 10. Higher rank is better.

tion 3.4. Apart from these proposed methods, we consider as baselines a battery of existing methods designed for batched black-box sequence design tasks: (1) Random, a method that randomly select proposal sequences at every round; (2) FB-VAE (Gupta & Zou, 2019) depicted in Section 3.1; (3) Evolution based (Brindle, 1980; Real et al., 2019) sequence design algorithm; and (4) DbAS (Brookes & Listgarten, 2018), described in Section 3.1.

We evaluate these sequence design algorithms by the average score of the top-10 and top-100 sequences in the resulting dataset $\mathcal{D}$ at each round. We plot the average score curves with regard to the number of rounds. We also use the area under the curve as a scalar metric for sample efficiency to compare the methods being evaluated. Specifically, since the area depends on the choice of x-axis, we simply cumulate the scores of all rounds to calculate the area. The result could be non-positive, since the score can take negative values.

## 4.2 RESULTS

**Transcription factor binding sites (TfBind)**. Protein sequences that bind with DNA sequences to adjust their activity are called transcription factors. In Barrera et al. (2016), the authors measure the binding properties between a battery of transcription factors and all possible length-8 DNA sequences through biological experiments. Concretely, we choose 15 transcription factors to serve as 15 different tasks. For each transcription factor, the algorithm needs to search for sequences that maximize the corresponding binding activity score. The size of the search space is $|\mathcal{V}|^L = 4^8 = 65536$. The number of total rounds is fixed to 10. For validation, we follow Angermüller et al. (2020b) and use one task (ZNF200_S265Y_R1) for hyperparameter selection. Then we test the algorithms' performance on the other 14 held-out tasks.

Figure 1 displays a comparison for all ten methods on three of the chosen binding affinity tasks. We can observe that after 10 rounds, our proposed methods perform consistently better than the

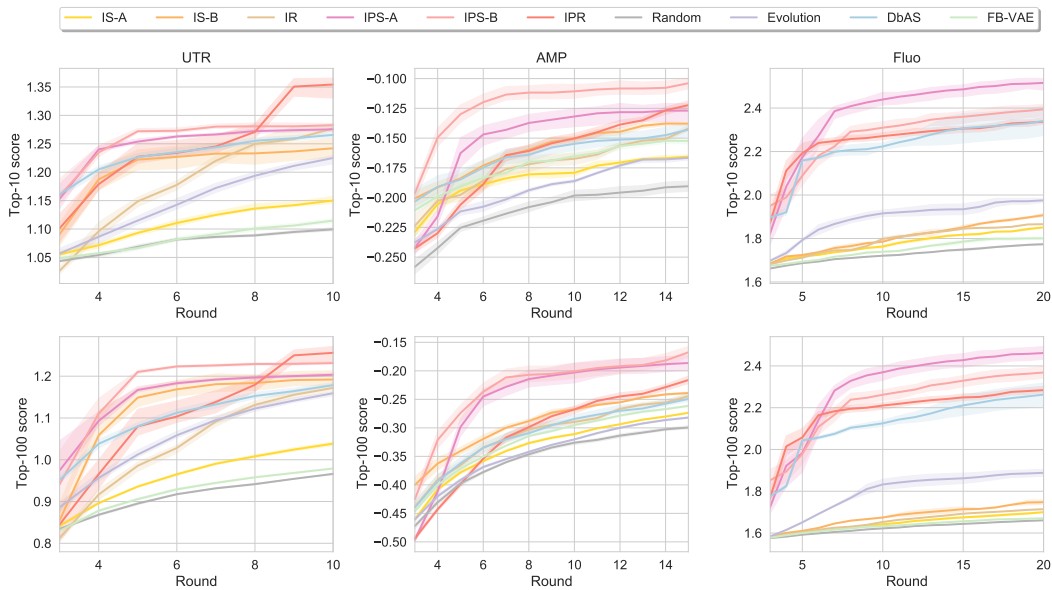

Figure 2: Top score (y-axis) curves of different methods on 3 sequence design problems (*left*: UTR, *middle*: AMP, *right*: Fluo) with regard to the number of rounds.

|      | IS-A  | IS-B  | IR    | IPS-A | IPS-B | IPR   | RANDOM | EVOLUTION | DbAS  | FB-VAE |
|------|-------|-------|-------|-------|-------|-------|--------|-----------|-------|--------|
| UTR  | 8.46  | 9.61  | 9.03  | 10.04 | 10.19 | 9.52  | 8.12   | 9.23      | 9.59  | 8.20   |
| AMP  | −5.67 | −5.11 | −5.37 | −4.65 | −4.39 | −5.42 | −5.96  | −5.79     | −5.39 | −5.54  |
| FLUO | 32.76 | 33.33 | 32.95 | 44.51 | 43.13 | 42.42 | 31.64  | 35.44     | 41.40 | 32.52  |

Table 3: Comparison of the area under top-100 curves for UTR, AMP and Fluo benchmarks. Larger area means better sample efficiency.

backward modeling methods like DbAS and FB-VAE in terms of both Top-10 and Top-100 scores. Among all baselines, the evolution method is the strongest one, and it beats IR on some of the tasks (see complete results in Table 4 and 5 in Appendix). We also find that IS-A and IS-B increase top scores slightly faster than other methods, especially on PBX4_REF_R2 and CRX_E80A_R1. This indicates that composite methods' way of using parameterized models to replace computational procedures is not the optimal solution for small-scale tasks.

**5' untranslated regions (UTR)**. The translation efficiency is mainly determined by the sequence of 5' UTR (Alipanahi et al., 2015). In Sample et al. (2019), the authors create a library of gene sequences with ribosome loading level as labels. They further train a convolutional neural network with this library to predict the relationship between a 5'UTR sequence and the corresponding gene expression level. We use this neural network as an oracle for this benchmark. The length of the gene sequences is fixed to 50, and thus the size of the search space is $4^{50}$. For this 5'UTR benchmark, we also allow each algorithm to explore for 10 rounds. Figure 2 (left) shows that our proposed composite methods significantly outperform other methods on the UTR task. Different from the results on the TfBind task, forward modeling methods do not achieve the best performance.

**Antimicrobial peptides (AMP)**. Protein modeling has recently become a popular sub-area of machine learning research. We are tasked to generate AMP sequences, which are short protein sequences against multi-resistant pathogens. We train a binary classifier model to classify whether a short protein sequence belongs to AMP and defer the related details to Appendix. This is the only task we consider regarding sequence design with alterable lengths, where the length of sequences ranges from 12 to 60. Since each entry of protein sequence has $|\mathcal{V}| = 20$ different choices on amino acids, the size of search space is $\sum_{L=12}^{60} 20^L$. We set the number of total rounds to be 15 for this task. Figure 2 (middle) clearly shows that the performances of IPS-A and IPS-B dominate the AMP generation task, which demonstrates the effectiveness of our composite strategy.

**Fluorescence proteins (Fluo)**. As another protein engineering task, we consider the optimization over fluorescent proteins, which is a commonly used test bed of modern molecular biology. This task is similar to the AMP task introduced above, but its ground-truth measurement relies on a regressor. We use a pretrained model taken from Rao et al. (2019) to act as our task oracle, which is a regressor trained to predict log-fluorescence intensity value over approximately 52,000 protein sequences of length 238 (Sarkisyan et al., 2016). More concretely, the regressor is fit on a small neighborhood of parent green fluorescent protein, and is then evaluated on a more distant protein. The training data is derived from the naturally occurring GFP in *Aequorea victoria*. The task Fluo's search space is $20^{238}$ and we set the number of rounds to 20. We demonstrate the Fluo results in Figure 2 (right), where we can see both composite methods and DbAS achieve much better results than other approaches. This might signify that for long sequence design tasks, backward modeling is superior to other modeling methods, which is not consistent with LFI (Papamakarios et al., 2019). We also summarize the sample efficiency results for the latter 3 benchmarks in Table 3 and 6, from which we can see that our proposed three composite methods perform remarkably promising results.

## 5 CONCLUSION

We propose a probabilistic framework that unifies likelihood-free inference and black-box optimization for designing biological sequences. This unified perspective enables us to design a variety of novel composite probabilistic sequence design methods combining the best of both worlds. Extensive experiments have demonstrated the benefits of the unified perspective. While the composite probabilistic methods usually outperform other baseline methods in most sequence design tasks we consider in this work, the key contribution of our paper is not just about the superiority of those composite methods, as different specific tasks might prefer different algorithmic configurations due to no free lunch theorem (Wolpert & Macready, 1997). Actually, we would like to attribute the strong performance to the unified probabilistic framework, which enables us to develop a richer algorithm pool, based on which we can design performant algorithms for particular sequence design tasks.

## ACKNOWLEDGEMENT

The authors would like to thank Christof Angermueller, Yanzhi Chen, Michael Gutmann, and anonymous reviewers for helpful feedbacks. Jie Fu thanks Microsoft Research Montreal for funding his postdoctoral position at University of Montreal and Mila. Yoshua Bengio acknowledges the funding from CIFAR, Samsung, IBM and Microsoft. Aaron Courville thanks the support of Samsung, Hitachi and CIFAR.

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

# A  MORE ABOUT METHODOLOGY

## A.1  BACKWARD MODELING OF THE MECHANISM

**About SMC-ABC**. In Algorithm 1, when we pick the top-$m$ $\boldsymbol{\theta}$ at the $r$-th round, the picked parameters actually follow such a distribution

$$p(\boldsymbol{\theta}|\boldsymbol{x}_o) \propto \sum_{l=1}^{r} p_l(\boldsymbol{\theta}) \cdot p(\|\boldsymbol{x} - \boldsymbol{x}_o\| < \epsilon \mid \boldsymbol{\theta}) = \left(\sum_{l=1}^{r} p_l(\boldsymbol{\theta})\right) \cdot p(\|\boldsymbol{x} - \boldsymbol{x}_o\| < \epsilon \mid \boldsymbol{\theta}) \quad (5)$$

where the $\epsilon$ here is implicitly defined by how "top" the selection process is, namely the ratio $\frac{m}{nr}$. As a result, we point out that in Algorithm 1, another more complicated form of the last step $p_{r+1}(\boldsymbol{\theta}) \leftarrow q_\phi(\boldsymbol{\theta})$ is $p_{r+1}(\boldsymbol{\theta}) \propto q_\phi(\boldsymbol{\theta})p(\boldsymbol{\theta})/\sum_l^r p_l(\boldsymbol{\theta})$, where an additional renormalizing term is involved. We ignore this term and used the simpler alternative in order to keep our main text clean. We refer interested readers to Beaumont et al. (2009) for more details.

**About Sequential Neural Posterior**. Define $p(\mathbf{x}) = \int p(\mathbf{x}|\boldsymbol{\theta})p(\boldsymbol{\theta})d\boldsymbol{\theta}$ and $\tilde{p}(\mathbf{x}) = \int p(\mathbf{x}|\boldsymbol{\theta})\tilde{p}(\boldsymbol{\theta})d\boldsymbol{\theta}$ for any arbitrary proposal distribution $\tilde{p}(\boldsymbol{\theta})$ which is not necessary to be the prior $p(\boldsymbol{\theta})$. What's more, we define $p(\boldsymbol{\theta}|\mathbf{x}) = p(\mathbf{x}|\boldsymbol{\theta})p(\boldsymbol{\theta})/p(\mathbf{x})$ and $\tilde{p}(\boldsymbol{\theta}|\mathbf{x}) = p(\mathbf{x}|\boldsymbol{\theta})\tilde{p}(\boldsymbol{\theta})/\tilde{p}(\mathbf{x})$ to be the true posterior and the proposal posterior.

Starting from the goal of approximating the true posterior,

$$\underset{q}{\arg\min}\, \mathbb{E}_{p(\mathbf{x})}[D_{\mathrm{KL}}(p(\boldsymbol{\theta}|\mathbf{x})\|q(\boldsymbol{\theta}|\mathbf{x}))] = \underset{q}{\arg\max} \int p(\mathbf{x})d\mathbf{x} \int p(\boldsymbol{\theta}|\mathbf{x}) \log q(\boldsymbol{\theta}|\mathbf{x})d\boldsymbol{\theta}$$

$$= \underset{q}{\arg\max} \int p(\boldsymbol{\theta}, \mathbf{x}) \log q(\boldsymbol{\theta}|\mathbf{x})d\boldsymbol{\theta}\mathbf{x}$$

$$= \underset{q}{\arg\max}\, \mathbb{E}_{p(\boldsymbol{\theta},\mathbf{x})}[\log q(\boldsymbol{\theta}|\mathbf{x})].$$

It seems that we can directly train the parameterized neural density estimator $q_\phi$ in a data driven manner via $\max_\phi \sum_i \log q_\phi(\boldsymbol{\theta}_i|\mathbf{x}_i)$ where $i$ is the index for data sample. When the number of training samples as well as the parameterization family of $\phi$ are large enough, the obtained $q_\phi$ would be close enough to the true posterior. This would require the data samples to follow $(\boldsymbol{\theta}_i, \mathbf{x}_i) \sim p(\boldsymbol{\theta}, \mathbf{x}) = p(\boldsymbol{\theta})p(\mathbf{x}|\boldsymbol{\theta})$. However, practically one uses a proposal $\tilde{p}(\boldsymbol{\theta})$ to first generate some $\{\boldsymbol{\theta}_i\}_i$ and then generate $\{\mathbf{x}_i\}_i$ by simulation. When the proposal distribution $\tilde{p}(\boldsymbol{\theta})$ is not exactly the prior distribution $p(\boldsymbol{\theta})$, the resulting $q_\phi(\cdot|\cdot)$ would be:

$$\tilde{p}(\boldsymbol{\theta}|\mathbf{x}) = p(\boldsymbol{\theta}|\mathbf{x})\frac{\tilde{p}(\boldsymbol{\theta})p(\mathbf{x})}{p(\boldsymbol{\theta})\tilde{p}(\mathbf{x})} \propto p(\boldsymbol{\theta}|\mathbf{x})\frac{\tilde{p}(\boldsymbol{\theta})}{p(\boldsymbol{\theta})}, \quad (6)$$

which is a biased estimation and is not what we want.

Three variants of SNP take different approaches to try to fix this bias. SNP-A (Papamakarios & Murray, 2016) first fits the biased proposal posterior $\tilde{p}(\boldsymbol{\theta}|\mathbf{x})$ in the aforementioned way and utilize the relation in Eq. 6 to solve for an unbiased estimation. This approach is restricted to mixture of Gaussian distribution family and thus has limited expressiveness. SNP-B (Lueckmann et al., 2017) uses importance sampling to address this issue via $\max_\phi \mathbb{E}_{(\mathbf{x},\boldsymbol{\theta})\sim p(\mathbf{x}|\boldsymbol{\theta})\tilde{p}(\boldsymbol{\theta})} \left[\frac{p(\boldsymbol{\theta})}{\tilde{p}(\boldsymbol{\theta})} \log q_\phi(\boldsymbol{\theta} \mid \mathbf{x})\right]$. One downside of this approach is the high variance involved by the importance weights $p(\boldsymbol{\theta})/\tilde{p}(\boldsymbol{\theta})$. SNP-C (Greenberg et al., 2019) proposes to use reparameterize the proposal posterior by setting $\tilde{q}_\phi(\boldsymbol{\theta}|\mathbf{x}) = q_\phi(\boldsymbol{\theta}|\mathbf{x})\frac{\tilde{p}(\boldsymbol{\theta})}{p(\boldsymbol{\theta})}\frac{1}{Z_\phi(\mathbf{x})}$ where $Z_\phi(\mathbf{x}) = \int q_\phi(\boldsymbol{\theta}|\mathbf{x})\frac{\tilde{p}(\boldsymbol{\theta})}{p(\boldsymbol{\theta})}d\boldsymbol{\theta}$ is the corresponding normalizing factor for $\mathbf{x}$. SNP-C then maximizes $\mathbb{E}_{(\mathbf{x},\boldsymbol{\theta})\sim p(\mathbf{x}|\boldsymbol{\theta})\tilde{p}(\boldsymbol{\theta})} [\log \tilde{q}_\phi(\boldsymbol{\theta} \mid \mathbf{x})]$.

**About Design by Adaptive Sampling**. We aim to approximate the posterior via minimizing the KL divergence:

$$\arg\min_q D_{\mathrm{KL}}(p(\mathbf{m}|\mathcal{E})\|q(\mathbf{m})) = \arg\max_q \int p(\mathbf{m}|\mathcal{E})\log q(\mathbf{m})d\mathbf{m}$$

$$= \arg\max_q \int p(\mathcal{E}|\mathbf{m})p(\mathbf{m})\log q(\mathbf{m})d\mathbf{m}$$

$$= \arg\max_q \mathbb{E}_{\tilde{q}(\mathbf{m})}\left[\frac{p(\mathbf{m})}{\tilde{q}(\mathbf{m})}p(\mathcal{E}|\mathbf{m})\log q(\mathbf{m})\right],$$

where $\tilde{q}(\mathbf{m})$ could be any distribution of $\mathbf{m}$. Brookes et al. (2019) takes this formulation. Brookes & Listgarten (2018) only differs in the place that it ignores the denominator term. According to Angermüller et al. (2020b), we choose the latter variant as one of our baselines because it is more stable in practice. We refer interested readers to Brookes & Listgarten (2018) for more details.

Notice that we are not doing exactly the same things for LFI and black-box sequence design. Since LFI models flexible posterior $p(\boldsymbol{\theta}|\mathbf{x})$ which is a distribution for arbitrary $\mathbf{x}$, we can also choose to model $p(\mathbf{m}|s)$ for arbitrary $s$. Nevertheless, in the neural network modeling, conditioning by a scalar value is not an effective approach as the effect of low dimensional scalar value conditioning may be covered by other high dimensional input. Therefore, we choose to directly model the target posterior $p(\mathbf{m}|\mathcal{E})$ with a single neural network.

## A.2 FORWARD MODELING OF THE MECHANISM

We still use $\tilde{p}(\boldsymbol{\theta})$ to denote an arbitrary proposal distribution and $\tilde{p}(\boldsymbol{\theta},\mathbf{x}) := p(\mathbf{x}|\boldsymbol{\theta})\tilde{p}(\boldsymbol{\theta})$. Then we have

$$\arg\min_q \mathbb{E}_{\tilde{p}(\boldsymbol{\theta})}\left[D_{\mathrm{KL}}\left(p(\mathbf{x}|\boldsymbol{\theta})\|q(\mathbf{x}|\boldsymbol{\theta})\right)\right] = \arg\max_q \int \tilde{p}(\boldsymbol{\theta})d\boldsymbol{\theta}\int p(\mathbf{x}|\boldsymbol{\theta})\log q(\mathbf{x}|\boldsymbol{\theta})d\mathbf{x}$$

$$= \arg\max_q \mathbb{E}_{\tilde{p}(\boldsymbol{\theta},\mathbf{x})}\left[\log q(\mathbf{x}|\boldsymbol{\theta})\right].$$

We point out that with much enough data and large enough expressiveness of the neural density estimator parameterization family, no matter what proposal $\tilde{p}(\boldsymbol{\theta})$ is used to provide training samples $\{(\boldsymbol{\theta}_i,\mathbf{x}_i)\}_i \sim \tilde{p}(\boldsymbol{\theta},\mathbf{x})$, we have the resulting $q_{\hat{\phi}}(\boldsymbol{\theta}|\mathbf{x})$ equals true likelihood $p(\mathbf{x}|\boldsymbol{\theta})$ in the support of the proposal. What SNL gives is an unbiased estimation and thus does not have the same problem as SNP.

Now we elaborate the construction of $\tilde{q}(\mathbf{m})$ in Iterative Scoring algorithm. Here we use the notation $\tilde{q}(\mathbf{m})$ to denote our *approximation* of the posterior $p(\mathbf{m}|\mathcal{E})$. Notice that we want $\tilde{q}(\mathbf{m}) \propto p(\mathbf{m}) \cdot p(\mathcal{E}|\mathbf{m})$. If we choose Example A to serve as the definition of event $\mathcal{E}$, then the samples of $\tilde{q}(\mathbf{m})$ can be obtained in this way: (1) sample $\mathbf{m}$ from prior $p(\mathbf{m})$ and (2) accept this sample if $\hat{f}_\phi(\mathbf{m})$ is larger than threshold $s$, or otherwise reject it. Alternatively, if we choose Example B, we have $\tilde{q}(\mathbf{m}) \propto p(\mathbf{m}) \cdot \exp(\hat{f}_\phi(\mathbf{m})/\tau)$. Similar to SNL, we do MCMC sampling from this unnormalized probability function.

## A.3 MODELING A PROBABILITY RATIO

**About Sequential Neural Ratio**. Dataset $\mathcal{D}$ is generated in the way that (1) first sample $\boldsymbol{\theta} \sim p(\boldsymbol{\theta})$ and (2) simulate $\mathbf{x} \sim p(\mathbf{x}|\boldsymbol{\theta})$. Consequently, $\mathcal{D}$ follows the distribution $p(\boldsymbol{\theta})p(\mathbf{x}|\boldsymbol{\theta}) = p(\boldsymbol{\theta},\mathbf{x})$. On the other hand, the other dataset $\mathcal{D}'$ generates $\boldsymbol{\theta}$ and $\mathbf{x}$ in parallel and independent manner. Notice here $\boldsymbol{\theta} \sim p(\boldsymbol{\theta})$ and $\mathbf{x}$ follows the marginal distribution: $\mathbf{x} \sim p(\mathbf{x}) = \int p(\boldsymbol{\theta})p(\mathbf{x}|\boldsymbol{\theta})d\boldsymbol{\theta}$.

**Proof of Proposition 1**.

*Proof.* We define a functional $\mathcal{F}$ to be the optimization objective:

$$\mathcal{F}[d] = \mathbb{E}_{\mathbf{a}\sim p_0(\mathbf{a})}[\log d(\mathbf{a})] + \mathbb{E}_{\mathbf{a}\sim p_1(\mathbf{a})}[\log(1-d(\mathbf{a}))]$$

We calculate its functional derivative. For arbitrary function $u$ and infinite small $\epsilon$

$$\mathcal{F}[d + \epsilon u] - \mathcal{F}[d] = \mathbb{E}_{p_0}[\log(1 + \epsilon \frac{u}{d})] + \mathbb{E}_{p_1}[\log(1 + \epsilon \frac{-u}{1-d})]$$

$$= \epsilon \int u \cdot \left( \frac{p_0}{d} + \frac{-p_1}{1-d} \right) + \mathcal{O}(\epsilon)$$

$$\Rightarrow \lim_{\epsilon \to 0} \frac{\mathcal{F}[d + \epsilon u] - \mathcal{F}[d]}{\epsilon} = \int u \cdot \left( \frac{p_0}{d} + \frac{-p_1}{1-d} \right) = \int u \cdot \delta\mathcal{F}.$$

We set the functional derivative to zero:

$$\delta\mathcal{F} = 0 \Rightarrow \frac{p_0}{p_1} = \frac{d}{1-d}$$

$$\Rightarrow d(\mathbf{a}) = \frac{p_0(\mathbf{a})}{p_0(\mathbf{a}) + p_1(\mathbf{a})}.$$

This optimal function $d^*$ apparently takes value in $[0, 1]$. □

**About Iterative Ratio**. Notice that in Algorithm 8 we construct two datasets: $\tilde{\mathcal{D}}$ and $\tilde{\mathcal{D}}'$. To generate $\tilde{\mathcal{D}}$, we need to be able to pick some sequence samples $\mathbf{m}$ from $\mathcal{D}$ and make the selected ones follow the posterior $p(\mathbf{m}|\mathcal{E})$. This procedure will depend on our choice of event $\mathcal{E}$. For Example A this is easy, since we just need to filter out the sequences whose oracle value is smaller than the threshold. However, for Example B, it is hard to do similar things, since given score value from $\mathcal{D}$ we only know the unnormalized value of posterior probability, and cannot determine which sequence should be filtered out. The construction of $\tilde{\mathcal{D}}'$ which follows prior distribution $p(\mathbf{m})$ is trivial.

### A.4 COMPOSITE PROBABILISTIC METHODS

For IPS, the optimization with regard to the second parameterized model $q_\psi(\mathbf{m})$ is

$$q_\psi = \arg\min_q D_{\mathrm{KL}}(\tilde{q}(\mathbf{m})\|q) = \arg\max_q \int \tilde{q}(\mathbf{m}) \log q(\mathbf{m}) d\mathbf{m}$$

$$= \arg\max_q \int p(\mathbf{m}|\mathcal{E}) \log q(\mathbf{m}) d\mathbf{m} = \arg\max_q \int p(\mathcal{E}|\mathbf{m}) p(\mathbf{m}) \log q(\mathbf{m}) d\mathbf{m}.$$

The exact value of $p(\mathcal{E}|\mathbf{m})$ depends on different choices of configuration feature $\mathcal{E}$ in Section 2.2. On the other hand, IPR, like IR, also only has one variant, which is with Example A:

$$q_\psi = \arg\min_q D_{\mathrm{KL}}(\tilde{q}(\mathbf{m})\|q) = \arg\max_q \int r_\phi(\mathbf{m}) p(\mathbf{m}) \log q(\mathbf{m}) d\mathbf{m},$$

which is a tractable optimization problem. Both IPR and IR are not fit for Example B since it cannot provide an exact probability value and thus cannot be adopted to construct $\tilde{\mathcal{D}}$.

## B MORE ABOUT EXPERIMENTS

Random method uses no neural network model. FB-VAE uses a VAE model. The encoder of the VAE first linearly transform one-hot input into a hidden feature which is 64 dimension, and then separately linearly transform to a 64-dimension mean output and 64-dimension variance output. The decoder contains a $64 \times 64$ linear layer and a linear layer that maps the hidden feature to categorical output. All other methods utilize bi-directional long short-term memory model (BiLSTM) (Hochreiter & Schmidhuber, 1997) with a linear embedding layer. Both the embedding dimension and the hidden size of LSTM is set to 32. For composite methods that use two models, we use one-layer LSTM for each of them. For the other algorithms that only use one LSTM, we set its number of

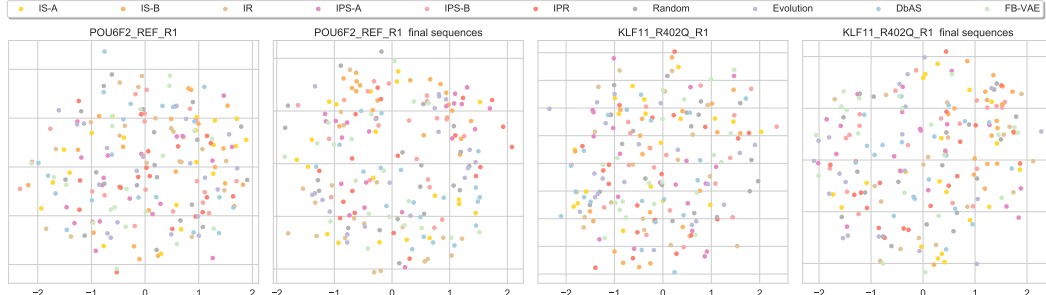

Figure 3: Diversity visualization results for two TfBind tasks.

|  | IS-A | IS-B | IR | IPS-A | IPS-B | IPR | RANDOM | EVO. | DBAS | FB-VAE |
|---|---|---|---|---|---|---|---|---|---|---|
| POU6F2_REF_R1 | 9 | 7 | 3 | 5 | 10 | 8 | 1 | 4 | 6 | 2 |
| KLF11_R402Q_R1 | 8 | 6 | 5 | 7 | 9 | 10 | 1 | 4 | 2 | 3 |
| EGR2_R359W_R1 | 4 | 5 | 8 | 9 | 7 | 6 | 1 | 2 | 10 | 3 |
| HOXD13_S316C_R1 | 8 | 10 | 3 | 4 | 7 | 9 | 1 | 5 | 6 | 2 |
| HOXB7_K191R_R1 | 10 | 8 | 6 | 4 | 9 | 3 | 1 | 5 | 7 | 2 |
| PBX4_REF_R2 | 10 | 9 | 8 | 7 | 5 | 4 | 1 | 6 | 2 | 3 |
| GFI1B_A204T_R1 | 8 | 7 | 3 | 4 | 9 | 10 | 1 | 6 | 5 | 2 |
| FOXC1_REF_R1 | 10 | 8 | 3 | 5 | 7 | 9 | 1 | 4 | 6 | 2 |
| KLF1_REF_R1 | 6 | 4 | 7 | 9 | 8 | 10 | 3 | 1 | 2 | 5 |
| SIX6_REF_R1 | 8 | 7 | 3 | 10 | 9 | 6 | 1 | 5 | 2 | 4 |
| ARX_L343Q_R2 | 10 | 3 | 4 | 7 | 9 | 5 | 1 | 6 | 8 | 2 |
| CRX_E80A_R1 | 9 | 10 | 7 | 5 | 8 | 2 | 1 | 6 | 3 | 4 |
| ESX1_K193R_R1 | 9 | 6 | 3 | 5 | 10 | 8 | 1 | 4 | 7 | 2 |
| VSX1_G160D_R1 | 9 | 7 | 4 | 6 | 10 | 8 | 1 | 3 | 5 | 2 |
| AVERAGE | 8.43 | 6.93 | 4.79 | 6.21 | 8.36 | 7.00 | 1.14 | 4.36 | 5.07 | 2.71 |

Table 4: Top-10 score ranking for the TfBind instances that we adopt. "Evo." stands for the evolution algorithm.

layers to be two. No Dropout (Srivastava et al., 2014) is used in LSTM models. In this way, the number of parameters of the VAE is slightly larger than that of the two layer BiLSTM, and all methods (except Random) share similar model parameter size.

All the experiments are repeated with fifty random seeds and report the mean value (and also standard deviation in the figure plots). We set $p(\mathbf{m})$ to be uniform prior for all tasks for simplicity, which uniformly samples from the dictionary $\mathcal{V}$ for each entry of the sequence. For length alterable task, we first uniformly sample the length between minimum length and maximum length and then sample each entry.

We explain details about evolution based method mentioned in the main text, which can be seen as a substantial example of directed evolution (Chen & Arnold, 1991). Like other model based methods, Evolution also trains an LSTM regressor to predict the score of a sequence, which is further used to assist in the reproduce procedure. The Evolution algorithm maintains a generation list through the whole exploration process. In each round, the method mutates and reproduces the sequences to enlarge the generation list, and then utilizes the learned regressor to select top sequences for the next generation.

For validation, we follow (Angermüller et al., 2020b) and sweep each algorithm for fifty trials and pick the best configuration. We tune learning rate and whether to re-initialize the optimizer for each new round for all methods. We tune threshold for DbAS, FB-VAE and the methods that is with Example A. For the other choice of $\mathcal{E}$, we tune the temperature. For evolution, we tune the number of offsprings for each sequence in generation list, the probability of substitution, insertion and deletion. For TfBind we use ZNF200_S265Y_R1 for validation. For UTR, AMP and Fluo, since we only have one oracle instance for each benchmark, we do not use a hold-out validation method.

| | IS-A | IS-B | IR | IPS-A | IPS-B | IPR | RANDOM | EVO. | DBAS | FB-VAE |
|---|---|---|---|---|---|---|---|---|---|---|
| POU6F2_REF_R1 | 10 | 8 | 3 | 5 | 9 | 6 | 1 | 7 | 4 | 2 |
| KLF11_R402Q_R1 | 10 | 7 | 4 | 8 | 9 | 6 | 1 | 5 | 3 | 2 |
| EGR2_R359W_R1 | 7 | 8 | 4 | 10 | 5 | 9 | 1 | 3 | 6 | 2 |
| HOXD13_S316C_R1 | 10 | 9 | 3 | 4 | 7 | 6 | 1 | 5 | 8 | 2 |
| HOXB7_K191R_R1 | 10 | 9 | 3 | 6 | 8 | 4 | 1 | 5 | 7 | 2 |
| PBX4_REF_R2 | 10 | 9 | 7 | 5 | 6 | 4 | 1 | 8 | 3 | 2 |
| GFI1B_A204T_R1 | 10 | 7 | 4 | 5 | 9 | 6 | 1 | 8 | 3 | 2 |
| FOXC1_REF_R1 | 10 | 7 | 4 | 8 | 6 | 9 | 1 | 5 | 3 | 2 |
| KLF1_REF_R1 | 8 | 6 | 5 | 9 | 7 | 10 | 1 | 4 | 3 | 2 |
| SIX6_REF_R1 | 9 | 7 | 5 | 10 | 8 | 4 | 1 | 6 | 3 | 2 |
| ARX_L343Q_R2 | 10 | 7 | 4 | 8 | 9 | 3 | 1 | 6 | 5 | 2 |
| CRX_E80A_R1 | 10 | 9 | 5 | 7 | 8 | 4 | 1 | 6 | 3 | 2 |
| ESX1_K193R_R1 | 10 | 9 | 3 | 4 | 8 | 6 | 1 | 5 | 7 | 2 |
| VSX1_G160D_R1 | 10 | 9 | 3 | 5 | 8 | 7 | 1 | 4 | 6 | 2 |
| AVERAGE | 9.57 | 7.93 | 4.07 | 6.71 | 7.64 | 6.00 | 1.00 | 5.50 | 4.57 | 2.00 |

Table 5: Top-100 score ranking for the TfBind instances that we adopt. "Evo." stands for the evolution algorithm.

| | IS-A | IS-B | IR | IPS-A | IPS-B | IPR | RANDOM | EVOLUTION | DBAS | FB-VAE |
|---|---|---|---|---|---|---|---|---|---|---|
| UTR | 10.87 | 11.71 | 11.43 | 12.06 | 12.15 | 11.94 | 10.60 | 11.20 | 11.89 | 10.65 |
| AMP | −2.98 | −2.67 | −2.84 | −2.54 | −2.16 | −2.74 | −3.36 | −3.09 | −2.73 | −2.80 |
| FLUO | 35.31 | 35.82 | 35.59 | 46.19 | 44.28 | 43.88 | 34.33 | 37.40 | 43.45 | 34.78 |

Table 6: Comparison of the area under top-10 curves for UTR, AMP and Fluo benchmarks. Larger area means better sample efficiency.

For TfBind benchmark, we use the following transcription factor instances and treat them as different black-box optimization tasks: ZNF200_S265Y_R1, POU6F2_REF_R1_8, KLF11_R402Q_R1, EGR2_R359W_R1, HOXD13_S316C_R1, HOXB7_K191R_R1, PBX4_REF_R2, GFI1B_A204T_R1, FOXC1_REF_R1, KLF1_REF_R1, SIX6_REF_R1, ARX_L343Q_R2, CRX_E80A_R1, ESX1_K193R_R1 and VSX1_G160D_R1. We do not do post-processing such as score normalization whitening for the data for simplicity. We first calculate the area under curve to summarize the performance in a scalar output, and put the ranking result for each algorithm in Table 4 and Table 5, which provide more details for Table 2. To further investigate the diversity of different algorithms, we choose two TfBind instances (POU6F2_REF_R1 and KLF11_R402Q_R1) and visualize the resulting sequences with T-SNE (van der Maaten & Hinton, 2008) in Figure 3. We provide two visualization views for both task instances: (1) we uniformly sample 20 sequences from the whole $n \cdot R$ sequences for each algorithm and visualize them; (2) for each algorithm, we visualize 20 sequences uniformly sampled from the last batch (*i.e.*, at the last round). This is notated with "final sequences" in the figure. We do not visualize all the sequences for simplicity. We use Hamming distance in the computation of T-SNE. From Figure 3, we can see that there is no obvious difference for the evaluated methods. This indicates that our proposed methods can achieve better performance while maintaining on-par diversity level with the baselines. This is not exactly consistent to the findings of Angermüller et al. (2020a), which claims some algorithms such as DbAS achieve very limited diversity. We do not use the "optima fraction" metric in Angermüller et al. (2020a;b), since this metric may not deal with multimode oracle landscape well and needs extra unstable computation such as clustering. Besides, this metric cannot generalize to other benchmarks.

We elaborate the construction of our AMP oracle. We use the AMP dataset from (Witten & Witten, 2019) which contains 6,760 AMP sequences. A multilayer perceptron classifier is trained to predict if a protein sequence can prohibit the growth of a particular pathogen in that AMP dataset. This classifier operates on the features extracted by ProtAlbert (Elnaggar et al., 2020) model. Following the setup in (Angermüller et al., 2020b), we treat the predicted logits as the ground-truth measurement. Moreover, we demonstrate the area under Top-10 curves for UTR, AMP and Fluo benchmarks in Table 6, which is a good complement for Table 3 but is missing due to limited space in the main text.

## C  RELATED WORKS AND DISCUSSION

**Likelihood-free inference**. We have already introduced the main classes of likelihood-free inference algorithms in the main text: (1) Approximate Bayesian Computation (ABC) method (Beaumont et al., 2009; Blum, 2009; Marin et al., 2012; Lintusaari et al., 2017) in Section 3.1; (2) Posterior modeling method that is also stated in Section 3.1, including classical ones (Tran et al., 2015; Li et al., 2017; Chen & Gutmann, 2019) and modern SNP methods (Papamakarios & Murray, 2016; Lueckmann et al., 2017; Greenberg et al., 2019); (3) Likelihood modeling method described in Section 3.2, also containing various classical algorithms (Wood, 2010; Mengersen et al., 2012; Drovandi et al., 2018) and modern SNL variants (Lueckmann et al., 2018; Papamakarios et al., 2019); and (4) Probability ratio modeling methods mentioned in Section 3.3 diverge in estimating likelihood ratio (Gutmann & Hyvärinen, 2010; Gutmann et al., 2018; Brehmer et al., 2020) or likelihood-to-evidence ratio (Thomas et al., 2016; Izbicki et al., 2014), where the latter paradigm is a good fit for LFI problem (Hermans et al., 2019). Besides, there are also works about how to construct low-dimensional summary statistics for LFI (Fearnhead & Prangle, 2012; Chan et al., 2018; Chen et al., 2021).

**Machine learning based drug design**. Generative modeling and discriminative modeling are two basic ways of thinking in machine learning. In literature for sequence design, generative modeling is also known as *cross entropy method*. This is a famous kind of design method that is close to our "backward modeling of the mechanism" approach. Cross entropy methods seek to solve an expectation maximization problem (*i.e.*, $\max_p \mathbb{E}_{p(\mathbf{m})}[f(\mathbf{m})]$) where the sequences follow a distribution $p$. This can also be related to simulated annealing, a large family of black-box optimization algorithm – the sequential neural posterior could be thought to maintain a distribution which is gradually becoming sharper to a delta distribution at the optimal value. On the other hand, we think of this as a way for modeling the posterior $p(\mathbf{m}|\mathcal{E})$ and develop corresponding analysis under the probabilistic framework, which is like a more accurate version of cross entropy method. Many related methods (including the ones stated in Section 3.1) train the distribution by likelihood maximization for sequences with large scores, or use some sort of reweighting to achieve similar effects (Rubinstein & Kroese, 2004; de Boer et al., 2005; Neil et al., 2018; Gupta & Zou, 2019; Brookes et al., 2019).

Discriminative modeling usually goes in a "model-based optimization" way (terminology from Angermüller et al. (2020a)), *i.e.*, use a discriminative model $\hat{f}(\mathbf{m})$ to fit the real oracle $f(\mathbf{m})$ and act as a surrogate for it. The surrogate model can replace the true oracle $f(\mathbf{m})$ which involves costly biological experiments. This corresponds to our "forward modeling of the mechanism" in Section 3.2. Bayesian optimization (Shahriari et al., 2016) is a classical example, which utilizes $\hat{f}$ (typically a Gaussian process model) to define an acquisition function to guide the exploration and exploitation. Many modern biochemical methods also belong to this category (Gómez-Bombarelli et al., 2018; Hashimoto et al., 2018; Yang et al., 2019; Wu et al., 2019; Sample et al., 2019; Liu et al., 2020).

There seems not much related literature about probability ratio estimation based method in this topic. On the other hand, Hashimoto et al. (2018) shares a classification based approach with IR but they differ on how to use the classifier. This algorithm utilizes the learned classifier to update the proposal with multiplicative weights algorithm, making the proposal to have large probability where the classifier logit is small. Other categories of drug design methods include evolution algorithms (Brindle, 1980; Wierstra et al., 2008; Salimans et al., 2017; Yoshikawa et al., 2018; Jensen, 2019; Real et al., 2019; Ahn et al., 2020) that search over the target space with genetic operators like insert, mutation, and crossover, and reinforcement learning (Guimaraes et al., 2017; Neil et al., 2018; Zhou et al., 2019; Shi et al., 2020; Angermüller et al., 2020b) which see the formation of a drug as a Markov decision process and train the policy to learn highly-rewarding drugs. We do not find other work that is similar to our probability ratio modeling approach (Section 3.3) from the literature, which we take as a novel contribution.

