# OpenReview forum: "Unifying Likelihood-free Inference with Black-box Optimization and Beyond"
_ICLR.cc/2022/Conference — ICLR 2022 Spotlight_

### Official Review · Reviewer_1dZH · 2021-10-27

**Correctness:** 3
**Technical Novelty And Significance:** 2
**Empirical Novelty And Significance:** 2
**Recommendation:** 6
**Confidence:** 4

**Main Review:**

# Major comments
1) The paper is missing a related works section with an overview of existing methods for sequence design and likelihood free inference, and how they relate to the methods that were introduced in this paper.

2) Section 3.1: FB-VAE and DbAs are both Estimation of Distribution Algorithms (EDA) based on a generative model. EDA is closely related to expectation maximization (https://arxiv.org/pdf/1905.10474.pdf). Please describe more clearly the differences and similarities of SNP, EDA, and EM.

3) Section 3.1: Please describe the differences between FB-VAE and DbAs more clearly. Both approaches update a VAE iteratively by fitting it on the top scoring sequences. What does 'top' mean? Are there differences in the 'fitting' of the VAE?

4) Section 3.2: IS is a discriminative approach for sequence design (or blackbox optimization) similar to (Bayesian) model-based optimization (MBO). Please describe the similarities between IS and MBO and differences (if there are any).

5) Section 3.2: Please describe what 'construct q(m) using f(m)' means. What is f(m) and q(m) in your experiments? How were they trained and which hyper-parameters were optimized?

6) Section 3.2: What are the differences between IS-A and IS-B?

7) Section 3.3: IR seems like a minor variation of IS that uses a classifier instead of a regressor as surrogate models, which is not new. Are there any other differences?

8) Section 3.3: Please describe which models you used for IR in your experiments and how sequences were generated.

9) Section 3.4: Please describe more clearly how IPS and IPR relate to (and differ from) existing design methods that combine generative and discriminative models, e.g. RL, GANs, or optimizing a surrogate model using DbAs/CbAs, for example.

10) Section 3.4: Please describe the differences between IPS-A and IPS-B more clearly, including 'differ in the detailed construction of the distribution q(m)'.

11) Experiments: Since the performance of algorithms can be sensitive to the batch size, I would like to see experiments with a different batch size than 100, e.g. small (1), medium (100) and large (500).

12) Experiments: Please compare to Bayesian Optimization (and RL if possible) using the same surrogate model as used for IPR/IPS and tuning hyper-parameters in the same way.

13) Experiments: Please describe which hyper-parameters you tuned and how they were tuned.

14) Experiments: Please describe what the boolean feature \mathcal{E} is in all or your experiments.

15) Experiments: Please describe the 'Evolution' baseline more clearly.

16) Experiments: Please motivate why you report the average reward of the top-10/100 sequences instead of just the maximum reward (top-1). The average can be maximized by reporting identical or very similar sequences. More important for practical applications is that the optimizer finds a diverse set of high-reward sequences as explained in Angermueller et al, who used additional diversity metrics to quantify this.

17) Experiments: How did you initialize the optimization? I would like to also see experiments that are initialized with a small set of labeled sequence (e.g. one or few parent sequences/homologs), which often exist in practice.


# Minor comments
Section 1, 'de novo biological sequence design': Describe which kind of sequence (DNA, RNA, protein, molecules represented as strings, ...?).

Section 2, 2nd paragraph: Please cite reviews (e.g. http://arxiv.org/abs/2106.05466, http://www.nature.com/articles/s41592-019-0496-6) of existing ML design methods instead of single papers (Ahn, Gottipati, ...).

Denoting 'm' as sequence and 's = f(m)' as the oracle function value is confusing since 's' is the first letter of *s*equence. I strongly suggest to use 's' to denote a sequence and, for example, 'y = f(s)' to denote the function value.

Your benchmark problems seem to be similar to the benchmark problems introduced in Angermueller et al (except for Flu). If this is the case, please describe that you reused the benchmark problems from Angermueller et al and describe possible differences. By referencing Angermueller et al instead of describing each benchmark problem in detail, you can also shorten the Experimental section.

Experiments (Flu). You hypothesize that backward modeling techniques perform better since sequences are long. Although sequences are long, there may be only a few variable positions while most positions are conserved. Since generative models can easily fit this conservation better they may perform well in this case since the optimization problem becomes trivial as only few variable positions are mutated.

**Summary Of The Paper:**

The paper relates likelihood free inference to methods for biological sequences design and proposes new sequence design methods based on this insight.

**Summary Of The Review:**

* The outlined relation between likelihood free inference and sequence design (blackbox optimization) is interesting; I am not aware of any existing papers with this insight. However, I am not sure about the impact of this insight on how sequences are designed in practice.
* It is not described clearly enough how the proposed methods differ from existing methods for sequence design such as Bayesian model-based optimization, GANs, or RL.
* Important details about the proposed methods and performed experiments are missing, which makes it hard to understand and assess.

---

> ### Author Response · Authors · 2021-11-13
> **Response to Reviewer 1dZH (Part 1)**
>
> Thank you for your insightful comments. Below, we provide responses to your comments. We hope we could address your concern such that we could receive a better score. For better demonstration, we group related questions together.
>
> **About the related work section.** We have a discussion in Section C about related work from sequence design and likelihood-free inference fields and their connection to our work.
>
> **About connection between CEM, EM and our probabilistic framework.** Thank you for pointing to this important reference [Brookes et al.]. Cross entropy method (CEM, [Rubinstein et al.]) might be another (more well-known) name for EDA — which refer to $\max_{q} \mathbb{E}_{q(m)}[f(m)]$. [Brookes et al.] shows that the optimization CEM could be interpreted as an EM algorithm. On the other hand, as we discussed in Section C, CEM could relate to the “backward modeling of the mechanism” approach under our framework. (SNP is the family of likelihood-free inference algorithm corresponding to this methodology.) Under our framework, the optimization target is $\arg\min_q KL(p(m|\mathcal{E})||q(m)) = \arg\max\int p(\mathcal{E}|m)p(m)\log q(m)dm$, which could be viewed as a more general form of the CEM target $\arg\max_q \int f(m)\log q(m)dm$ (derived in [Brookes et al.]) if a corresponding form of $\mathcal{E}$ is defined such that $p(\mathcal{E}|m) \propto f(m) / p(m)$. As a result, both EM and EDA (CEM) could be interpreted in our probabilistic framework. We shall add related discussion into our Section C in the final version.
>
> **About "model-based optimization" (MBO).** We are not very certain about what MBO refers to here. (1) On one hand, generally speaking, "model-based"  means one has a model to imitate the behavior of the oracle $f(m)$. As we discussed in Section C, MBO is actually another expression of "discriminative / forward modeling ideology", and IS-A/B are just two of the concrete examples in this algorithm family. (2) On the other hand, MBO could mean a specific algorithm. In [Angermueller et al.], the authors use MBO to refer to a model-based RL algorithm (which utilizes the model to give reward) and latent-space MBO to refer to [Gómez-Bombarelli et al.] (which trains a GP regressor on a VAE-based latent space and do gradient ascent; this method would additionally require gradient information of the oracle). In this way, IS-A/B, MBO and latent-space MBO are different specific algorithms that follow the "discriminative / forward modeling ideology" in our Section 3.2.
>
> **About the IS algorithm.** As we have discussed in Section A.2, we use the notation  $\tilde q(m)$ to denote our approximation of the posterior $p(m|\mathcal{E})$. Notice that we want  $\tilde q(m)\propto p(m)p(E|m)$. If we choose Example A to serve as the definition of event $\mathcal{E}$, then the samples of $\tilde q(m)$ can be obtained in this way: (1) sample $m$ from prior $p(m)$ and (2) accept this sample if $\hat f_\phi(m)$ is larger than threshold $s$, or otherwise reject it. Alternatively, if we choose Example B, we have $\tilde q(m)\propto p(m) \exp( \hat f_\phi(m)/\tau )$. This is also how IS-A and IS-B differ.
>
> **About the IR algorithm.** There seems to be a misunderstanding, as IR is NOT a minor variant of IS. There is a fundamental difference between the contrastive modeling [Gutmann et al.] (such as GAN [Goodfellow et al.]) and directly learning a regressor. For IR, the probability ratio could be learned from the theoretical result of proposition 1 and be further used for the construction of the target posterior. The model architecture that IR adopts is the same as other algorithms (IS, IPS, IPR) in our draft, which is a Bi-LSTM as the binary classification logit model as we describe in Section B. As we described in Section 3.3 and Section A.3, the generation of the sequences is through an mcmc process whose target distribution is $r(m)\cdot p(m)$.
>
> **About the composite algorithm family.**  Could you elaborate more on this question? We are not very sure why you claim RL and GAN are combining generative and discriminative models. For RL algorithm (we guess you are referring to [Angermueller et al. ICLR2020]), it is a policy that is being learned rather than a generative model. As for GAN [Goodfellow et al.], there is indeed both a generator and discriminator within, but in that case, these two models are playing a (possibly zero-sum) game until reaching equilibrium. In other words, the discriminator in GAN is a helper to achieve a good generator — it is an approach, not the purpose. However, both the generator and discriminator are important in our composite algorithms, and they are *collaborating* rather than *competing*. The difference between IPS-A and IPS-B also comes from the two definitions that we describe in Section 2.2 Section A.2 and Section A.4.

---

> > ### Author Response · Authors · 2021-11-13
> > **Response to Reviewer 1dZH (Part 2)**
> >
> > **About different query set size.** Thank you for suggesting this ablation study. Due to the time limit, we only conduct experiments on IPR on TfBind (KLF11_R402Q_R1) following the same protocol in our draft. We report the average with fifty random seeds. We set query batch size to 25, 50, 100, and 200 and compare their results. Here the x-axis denotes the number of queries having been made, and the y-axis denotes the query batch size for each round. The following table shows that a smaller query batch size is more sample efficient, which makes sense because it would allow more exploration times. Here "/" means value not applicable. We shall extend this ablation study into the final version of our draft.
> >
> > KLF11_R402Q_R1 top-10
> >
> > |    |  300   | 400   | 500   | 600   | 700   | 800   | 900   | 1000  |
> > |:-----------------:|:-----:|:-----:|:-----:|:-----:|:-----:|:-----:|:-----:|:-----:|
> > | 25             |  0.438 | 0.443 | 0.445 | 0.449 | 0.455 | 0.458 | 0.461 | 0.464 |
> > | 50             |  0.409 | 0.429 | 0.437 | 0.441 | 0.443 | 0.445 | 0.449 | 0.454 |
> > | 100            |  0.408 | 0.430 | 0.442 | 0.444 | 0.444 | 0.445 | 0.446 | 0.447 |
> > | 200            |   /     | 0.369 |   /    | 0.405 |   /    | 0.428 |   /    | 0.433 |
> >
> > KLF11_R402Q_R1 top-100
> >
> > |    |  300   | 400   | 500   | 600   | 700   | 800   | 900   | 1000  |
> > |:-----------------:|:-----:|:-----:|:-----:|:-----:|:-----:|:-----:|:-----:|:-----:|
> > | 25             | 0.346 | 0.370 | 0.382 | 0.388 | 0.392 | 0.397 | 0.399 | 0.403 |
> > | 50             | 0.317 | 0.361 | 0.379 | 0.394 | 0.400 | 0.405 | 0.409 | 0.411 |
> > |     100    |  0.262 | 0.333 | 0.364 | 0.379 | 0.392 | 0.401 | 0.405 | 0.408 |
> > | 200            |  /  | 0.239 |    /   | 0.328 |  /     | 0.376 |  /     | 0.398 |
> >
> >
> > **About top-k metric.** The usage of top-k score could be rationalized that top-1 score can be *unstable* and easy to be affected by outliers. We report top-k for better robustness and reasonable algorithm evaluation. For the diversity concern, we have a related discussion in Section B.  What's more, all of the metrics (top-1/10/100) are actually consistent with each other. We demonstrate this point by reporting the IPR's top-1 score performance for TfBind across three tasks (KLF11_R402Q_R1, PBX4_REF_R2, CRX_E80A_R1). We report the results from the 3rd round to the 10th round in order to keep consistent with Figure 1. We shall extend this ablation study into the final version of our draft.
> >
> > KLF11_R402Q_R1
> >
> > |       Round       |    3   |   4   |   5   |   6   |   7   |   8   | 9     |  10 |
> > |:-----------------:|:-----:|:-----:|:-----:|:-----:|:-----:|:-----:|:-----:|-------|
> > | IPR-top1 | 0.437 | 0.439 | 0.453 | 0.457 | 0.457 | 0.457 | 0.458 | 0.458 |
> > |     IPR-top10     |  0.408 | 0.430 | 0.442 | 0.444 | 0.444 | 0.445 | 0.446 | 0.447 |
> > |     IPR-top100    |  0.262 | 0.333 | 0.364 | 0.379 | 0.392 | 0.401 | 0.405 | 0.408 |
> >
> > PBX4_REF_R2
> >
> > |       Round       |    3   |   4   |   5   |   6   |   7   |   8   | 9     |  10 |
> > |:-----------------:|:-----:|:-----:|:-----:|:-----:|:-----:|:-----:|:-----:|:-----:|
> > | IPR-top1 | 0.424 | 0.441 | 0.452 | 0.459 | 0.470 | 0.481 | 0.481 | 0.484 |
> > |     IPR-top10     |  0.361 | 0.382 | 0.405 | 0.422 | 0.431 | 0.440 | 0.450 | 0.455 |
> > |     IPR-top100    | 0.249 | 0.291 | 0.318 | 0.336 | 0.351 | 0.366 | 0.378 | 0.389 |
> >
> > CRX_E80A_R1
> >
> > |       Round       |    3   |   4   |   5   |   6   |   7   |   8   | 9     |  10 |
> > |:-----------------:|:-----:|:-----:|:-----:|:-----:|:-----:|:-----:|:-----:|:-----:|
> > |  IPR-top1  |   0.482 | 0.485 | 0.481 | 0.493 | 0.493 | 0.495 | 0.497 | 0.503 |
> > |     IPR-top10     | 0.403 | 0.429 | 0.439 | 0.453 | 0.457 | 0.467 | 0.475 | 0.483 |
> > |     IPR-top100    | 0.256 | 0.301 | 0.324 | 0.355 | 0.371 | 0.389 | 0.404 | 0.421 |
> >
> > **About the choice of $\mathcal{E}$.** As we describe for each algorithm in Section 3 and Section A, we use Example A for IS-A, IR, IPS-A, and IPR, and Example B for IS-B and IPS-B.
> >
> > **About the Evolution baseline.** As we describe in Section B, can be seen as a substantial example of directed evolution [Chen & Arnold]. Like other model-based methods, Evolution also trains an LSTM regressor to predict the score of a sequence, which is further used to assist in the reproduce procedure. The Evolution algorithm maintains a generation list through the whole exploration process. In each round, the method mutates and reproduces the sequences to enlarge the generation list, and then utilizes the learned regressor to select top sequences for the next generation.

---

> > > ### Author Response · Authors · 2021-11-13
> > > **Response to Reviewer 1dZH (Part 3)**
> > >
> > > **About the hyperparameters tuning.** As we describe in Section B, we use ZNF200_S265Y_R1 as the hold-out validation set for TfBind, and do not use a hold-out validation method for the other three tasks since we do not have access to one. For all the algorithms, we use the sweep feature of the weights and biases platform ([https://wandb.ai/site](https://wandb.ai/site)) to run fifty trials for each algorithm on each task, and pick the best hyperparameters. We tune the learning rate and whether to re-initialize the optimizer for each new round for all methods. We tune the threshold for DbAS, FB-VAE, and the methods that are with Example A. For the other choice of E, we tune the temperature. For evolution, we tune the number of offsprings for each sequence in the generation list, the probability of substitution, insertion, and deletion.
> > >
> > > **About the initialization.** The algorithms start with a batch sampled from the prior distribution as you stated.
> > >
> > > **About Bayesian optimization (BO).** As we discussed in Section C, BO belongs to the discriminative modeling under our probabilistic framework. The modeling of BO heavily relies on Gaussian process, which is known to perform poorly in modern machine learning because of the poor generalization ability of the kernel method in high dimension settings. On the other hand, the proposed IS algorithm could actually be seen as a deep neural network version of BO, replacing the Gaussian process regressor with a neural network. The acquisition function in BO could balance the exploration and exploitation and here in our case, the temperature coefficient could do the same job.
> > >
> > > **About the neural network architectures.** As we discussed in Section B, FB-VAE uses a VAE model which is described in Section B. All other methods utilize bi-directional long short-term memory (BiLSTM) [Hochreiter & Schmidhuber] models with a linear embedding layer. Both the embedding dimension and the hidden size of LSTM are set to 32. For composite methods that use two models, we use one-layer LSTM for each of them. For the other algorithms that only use one LSTM, we set its number of layers to be two. No Dropout is used in LSTM models. In this way, the number of parameters of the VAE is slightly larger than that of the two-layer BiLSTM, and all methods (except Random) share similar model parameter sizes.
> > >
> > > **About the minor comments.** (1) Generally speaking, all kinds of sequences are doable, but in this paper, we explore DNA and protein. (2) Thanks for referring to the important reviews, we shall modify the second paragraph as you suggest. (3) We are NOT using the suite from [Angermueller et al.].  In fact, as far as we know [Angermueller et al.] is not open-sourced, so we construct our own benchmark. For example, we train our own AMP oracle as we describe in the last paragraph of Section B. (4) Thank you for the analysis for Fluo. We shall take your analysis into consideration and elaborate more in the final version of this draft.
> > >
> > > **References**
> > >
> > > Brookes, David H., Akosua Busia, Clara Fannjiang, Kevin P. Murphy and Jennifer Listgarten. “A view of estimation of distribution algorithms through the lens of expectation-maximization.” *Proceedings of the 2020 Genetic and Evolutionary Computation Conference Companion* (2020): n. pag.
> > >
> > > Rubinstein, Reuven Y.. “The Cross-Entropy Method for Combinatorial and Continuous Optimization.” *Methodology And Computing In Applied Probability* 1 (1999): 127-190.
> > >
> > > Angermueller, Christof, David Belanger, Andreea Gane, Zelda E. Mariet, David Dohan, Kevin Murphy, Lucy J. Colwell and D. Sculley. “Population-Based Black-Box Optimization for Biological Sequence Design.” *ICML* (2020).
> > >
> > > Gómez-Bombarelli, Rafael, David Kristjanson Duvenaud, José Miguel Hernández-Lobato, Jorge Aguilera-Iparraguirre, Timothy D. Hirzel, Ryan P. Adams and Alán Aspuru-Guzik. “Automatic Chemical Design Using a Data-Driven Continuous Representation of Molecules.” *ACS Central Science* 4 (2018): 268 - 276.
> > >
> > > Gutmann, Michael U. and Aapo Hyvärinen. “Noise-contrastive estimation: A new estimation principle for unnormalized statistical models.” *AISTATS* (2010).
> > >
> > > Goodfellow, Ian J., Jean Pouget-Abadie, Mehdi Mirza, Bing Xu, David Warde-Farley, Sherjil Ozair, Aaron C. Courville and Yoshua Bengio. “Generative Adversarial Nets.” *NIPS* (2014).
> > >
> > > Angermueller, Christof, David Dohan, David Belanger, Ramya Deshpande, Kevin Murphy and Lucy J. Colwell. “Model-based reinforcement learning for biological sequence design.” *ICLR* (2020).
> > >
> > > Chen, Keqin and Frances H. Arnold. “Enzyme Engineering for Nonaqueous Solvents: Random Mutagenesis to Enhance Activity of Subtilisin E in Polar Organic Media.” *Bio/Technology* 9 (1991): 1073-1077.

---

> ### Author Response · Authors · 2021-11-18
> **Hoping that our response could address your concern**
>
> We would appreciate it if you can let us know if our response has addressed your concern and thus improved your assessment of our work. We look forward to hearing from you!

---

> ### Author Response · Authors · 2021-11-20
> **Inquiry regarding an update to our response**
>
> Dear Reviewer,
>
> We are extremely appreciative of your endeavour for reviewing our paper, and as such, we try our best to offer a thorough rebuttal. As the deadline for the discussion session approaches and other reviewers have provided their thoughts, we humbly request your feedback on our detailed rebuttal response.

---

> > ### Comment · Reviewer_1dZH · 2021-11-24
> > **Increased score**
> >
> > I appreciate your comprehensive response to my review and addressing most of my comments. I have increased my rating to 'marginally above the acceptance threshold'. Unfortunately, I could not respond before the rebuttal deadline.
> >
> > I would appreciate if you could describe the differences or your proposed methods (in particular IR and iterative variants) and existing methods clearly *in the main text* (now in section C).
> >
> > I would also appreciate if could compare the performance *all methods* for different batch sizes for the UTR benchmark instead of TfBind8 since TfBind8 can be solved relatively easily. Methods are expected to be more sample efficient if the batch size is small. What is more important is to know how the relative ranking of methods changes for different batch sizes.
> >
> > I cannot follow your argument why you are reporting top-k scores instead of only the top-1 score. Why is it beneficial to propose the same (good) sequence multiple times or with few additional mutations that may not be correlated with the fitness function at all. What is more important is to know if methods keep on exploring other distinct optima after they found one optimum, which you did not examine.
> >
> > I suggest adding a conclusions section that briefly summarizes your experiments and highlights performance differences and when which method is expected to perform well.

---

### Official Review · Reviewer_5Uhg · 2021-11-02

**Correctness:** 4
**Technical Novelty And Significance:** 4
**Empirical Novelty And Significance:** 3
**Recommendation:** 10
**Confidence:** 4

**Main Review:**

I found the paper to be extremely clear and well written, providing an excellent review of both the LFI and black-box sequence design literatures and drawing clear, clean parallels between the two fields.  The proposed algorithms are all sensible and seem to work well on the empirical tasks, and the connection between the two fields seems like a fruitful area for further exploration (especially using the presented framework).  I have few comments below:

Typos:
- On p. 4 it is stated that "...to model the general posterior $p(\theta|\mathbf{x})$, which takes arbitrary $\theta$ and $\mathbf{x}$ as two inputs and outputs a distribution", but these models only take $\mathbf{x}$ as input and output the conditional distribution over $\theta$ (i.e., the posterior).
- There are a number of typos in the supplementary materials (appendix).  The meaning was always clear, but that document could use some thorough copy editing.

**Summary Of The Paper:**

In this paper the authors draw direct parallels between likelihood-free inference (LFI) and black-box sequence design.  This allows that authors to draw parallels between existing methods from the LFI and black-box sequence design literatures.  In a few cases there is no direct analog in the black-box sequence design literature for a given LFI algorithm, and so the authors are able to immediately propose such an algorithm.  The authors also present a number of "composite" methods that combine ideas from a number of these approaches.

**Summary Of The Review:**

The paper is clear and well-written and provides a very useful conceptual framework tying two subfields together, with sufficient empirical evidence to show real gains from this approach.

---

> ### Author Response · Authors · 2021-11-13
> **Response to Reviewer 5Uhg**
>
> Thank you for your positive assessment and insightful comments. Besides, thank you for pointing out the typo. We have corrected them as "... the general posterior $p(\theta|x)$, which takes arbitrary $x$ as input and outputs a conditional distribution over $\theta$". We will modify our Section 3.1 in this way in our final version.
>
> We are happy to answer any other questions you may have.

---

### Official Review · Reviewer_JsdK · 2021-11-02

**Correctness:** 4
**Technical Novelty And Significance:** 3
**Empirical Novelty And Significance:** Not applicable
**Recommendation:** 6
**Confidence:** 4

**Main Review:**

Can you please comment on the relationship between your work and 'Derivative free optimization via repeated classification' https://arxiv.org/abs/1804.03761?

The paper builds up a variety of optimization methods, building on a line of work in the LFI literature. This leads to a lot of approaches to compare, and there is no clear indication as to what practitioners should use in practice. What do you actually suggest people should use?

Along these lines, the paper has far too few details about the actual optimization approaches. For example, IS-A and IS-B seem to be some of your strongest methods and these require MCMC to sample new proposed sequences. There is no discussion of how this MCMC is done. Further, there is no discussion of neural network architectures, optimization methods, etc. The paper would be stronger if it just focused on one method and provided sufficient details for practitioners to actually use it.

I found this sentence in sec 4.2 very unsatisfying. Do you have any further insights about performance differences? " This indicates that composite methods’ way of using parameterized models to replace computational procedures is not the optimal solution for small-scale tasks."

I was surprised that the error bars were so small in all of your experiments, as I expect that the trajectory of an optimizer has high variance due, for example, to the initial set of sequences that are sampled. What do your error bars correspond to? Are they standard errors or standard deviations? Also, what sources of randomness are you accounting for when you generate multiple random trials?

For the methods that combine both a generative and discriminative models, there are two sources of approximation error. It would be very helpful if you isolated these by performing oracle experiments, where you assume that the forward model is exactly correct. This can be used to isolate the impact, for example, of using the amortized sampler q_\phi in Alg 10. Just replace the forward model with the ground truth objective function. Can you run a quick experiment?


**Summary Of The Paper:**

The paper draws on connections between likelihood-free inference and black-box optimization to propose new black-box optimization methods. In general, the goal here is to not find the exact optimum of the black-box objective, but to sample from a set of sequences with high-quality objective. This is akin to the problem of collecting posterior samples in a likelihood-free inference problem.

The paper provides a number of proposed methods and compares them on some benchmark sequence optimization problems that have appeared in recent literature.

**Summary Of The Review:**

The paper has some interesting methods, and the connection to LFI is helpful. However, it does not have a clear empirical recommendation for what algorithm readers should use going forward and the paper does not provide adequate details to understand how to go about actually using these methods in practice.

---

> ### Author Response · Authors · 2021-11-12
> **Response to Reviewer JsdK (Part 1)**
>
> Thank you for your insightful comments. There seem to be some misunderstanding as a result of our negligence in writing, and we hope our explanation below could help you address them.
>
> **About "repeated classification" [Hashimoto et al.] paper.** Thank you for suggesting this discussion. Among the six algorithms that our draft proposes, the algorithm in [Hashimoto et al.] is most close to the Iterative Ratio (IR) algorithm. With the definition of Example A, the $d_\phi(m)$ in IR algorithm is solving a classification problem between high-score sequences and low-score sequences, which is similar to the behavior of $h(\cdot)$ in algorithm 1 of [Hashimoto et al.]. Their differences emerge after the training of such a classifier: IR constructs the ratio between target posterior and prior distribution with learned $d_\phi(m)$ which is justified by the proposition 1 in our draft, and further uses the ratio for the proposal of next round; on the other hand, inspired by classical cutting-plane algorithms, [Hashimoto et al.] uses the learned classifier to update the proposal with multiplicative weights algorithm [Arora et al.], making the proposal to have large probability where $h(\cdot)$ is small. As of our Iterative Scoring (IS) algorithm, in IS a regressor instead of classifier is learned, and IS directly uses the regressor to propose new sequences, while  [Hashimoto et al.] uses the learned classifier to update proposal. We think it is also doable if one uses a regressor in the framework of multiplicative weights algorithm adopted by [Hashimoto et al.]. It is our negligence that we are not aware of this work when writing our draft, and we shall add more relevant discussion into our Section 3.3 and related work section in our final version.
>
> **Practitioner takeaway message.** In Section 4.2, the message we actually want to express is: for small-scale tasks, many proposed algorithms including composite algorithms perform approximately equally well and meanwhile, the composite methods are slightly more computational consuming than other simpler proposed methods. We do not claim that there exists any particular algorithm that would always outperform others under all settings, which is unreasonable due to the different properties of different tasks. On the other hand, we find composite methods could generally obtain SOTA results, especially in larger experiments which would be more realistic. We do not specifically distinguish within three composite methods either as we believe no one would consistently perform the best. In practical application, since we will face unknown tasks, we should still use composite methods as first trials because of their satisfying performance in numerical experiments. We shall rewrite relevant parts in the experiment section and avoid misleading arguments in our final version as you suggest.
>
> **References**
>
> Hashimoto, Tatsunori B., Steve Yadlowsky and John C. Duchi. “Derivative Free Optimization Via Repeated Classification.” *AISTATS* (2018).
>
> Arora, Sanjeev, Elad Hazan and Satyen Kale. “The Multiplicative Weights Update Method: a Meta-Algorithm and Applications.” *Theory Comput.* 8 (2012): 121-164.

---

> > ### Author Response · Authors · 2021-11-12
> > **Response to Reviewer JsdK (Part 2)**
> >
> > **Practice details.** Thank you for suggesting a more detailed experimental description. We put some relevant descriptions in Appendix but we agree more about implementation will definitely improve the quality of our draft. We shall make a more detailed one in the final version as you suggest. We also put relevant statement below. For neural network architectures, FB-VAE uses a VAE model. The encoder of the VAE first linearly transforms one-hot input into a hidden feature which is 64 dimension, and then separately linearly transforms to a 64-dimension mean output and 64-dimension variance output. The decoder contains a 64×64 linear layer and a linear layer that maps the hidden feature to categorical output. All other methods utilize bi-directional long short-term memory model (BiLSTM) [Hochreiter & Schmidhuber] with a linear embedding layer. Both the embedding dimension and the hidden size of LSTM is set to 32. For composite methods that use two models, we use one-layer LSTM for each of them. For the other algorithms that only use one LSTM, we set its number of layers to be two. No Dropout is used in LSTM models. In this way, the number of parameters of the VAE is slightly larger than that of the two-layer BiLSTM, and all methods (except Random) share similar model parameter sizes. For training methods, we use Adam [Kingma & Ba] optimizer. For sampling methods, we adopt the rejection sampling method which could be considered as a basic MCMC variant and we use the prior $p(m)$ as the proposal. As a side note, we want to claim that IS-A and IS-B are not the very strongest methods especially under larger / realistic settings. We will put a more specified version than the current statement in our final version.
> >
> > **About the error bars.** We have a description of random repeated runs in Section B. We perform fifty random trials and report the standard deviation. We sample an initial set of sequences from prior and stick to this fixed set. In our setting, the randomness source includes different initialization of neural networks, the sampling process from the proposal at each iteration, and stochastic gradients in the training. We shall add more relevant discussion in the final version. Actually, we think our randomness is consistent with the randomness scale in [Angermueller et al.], could you check the experimental part in that paper?
> >
> > **For impact isolation in composite methods.**
> > We perform this ablation experiment for IPR as you suggest. We replace the forward part with oracle and use "IPR-oracle" to refer to it. We test with TfBinding on KLF11_R402Q_R1, PBX4_REF_R2 and CRX_E80A_R1, and find IPR-oracle is consistently better than IPR as it exploits the extra information of oracle. We report the numbers in a way that the reviewer could have a comparison with the Figure 1 in our draft. Here are the results.
> >
> > KLF11_R402Q_R1
> >
> > |       Round       |    3   |   4   |   5   |   6   |   7   |   8   | 9     |  10 |
> > |:-----------------:|:-----:|:-----:|:-----:|:-----:|:-----:|:-----:|:-----:|-------|
> > |     IPR-top10     |  0.408 | 0.430 | 0.442 | 0.444 | 0.444 | 0.445 | 0.446 | 0.447 |
> > |  IPR-oracle-top10 |   0.432 | 0.434 | 0.439 | 0.449 | 0.454 | 0.462 | 0.473 | 0.487 |
> > |     IPR-top100    |  0.262 | 0.333 | 0.364 | 0.379 | 0.392 | 0.401 | 0.405 | 0.408 |
> > | IPR-oracle-top100 |  0.374 | 0.397 | 0.403 | 0.408 | 0.411 | 0.413 | 0.416 | 0.420 |
> >
> > PBX4_REF_R2
> >
> > |       Round       |    3   |   4   |   5   |   6   |   7   |   8   | 9     |  10 |
> > |:-----------------:|:-----:|:-----:|:-----:|:-----:|:-----:|:-----:|:-----:|:-----:|
> > |     IPR-top10     |  0.361 | 0.382 | 0.405 | 0.422 | 0.431 | 0.440 | 0.450 | 0.455 |
> > |  IPR-oracle-top10 |  0.431 | 0.449 | 0.466 | 0.474 | 0.478 | 0.480 | 0.482 | 0.483 |
> > |     IPR-top100    | 0.249 | 0.291 | 0.318 | 0.336 | 0.351 | 0.366 | 0.378 | 0.389 |
> > | IPR-oracle-top100 |  0.331 | 0.367 | 0.398 | 0.420 | 0.433 | 0.441 | 0.447 | 0.451 |
> >
> > CRX_E80A_R1
> >
> > |       Round       |    3   |   4   |   5   |   6   |   7   |   8   | 9     |  10 |
> > |:-----------------:|:-----:|:-----:|:-----:|:-----:|:-----:|:-----:|:-----:|:-----:|
> > |     IPR-top10     | 0.403 | 0.429 | 0.439 | 0.453 | 0.457 | 0.467 | 0.475 | 0.483 |
> > |  IPR-oracle-top10 | 0.459 | 0.488 | 0.495 | 0.496 | 0.496 | 0.496 | 0.496 | 0.497 |
> > |     IPR-top100    | 0.256 | 0.301 | 0.324 | 0.355 | 0.371 | 0.389 | 0.404 | 0.421 |
> > | IPR-oracle-top100 | 0.340 | 0.409 | 0.458 | 0.472 | 0.477 | 0.479 | 0.482 | 0.483 |
> >
> > We shall add more ablation in this isolation comparison into the draft in the final version.
> >
> > **References**
> >
> > Hochreiter, Sepp and Jürgen Schmidhuber. “Long Short-Term Memory.” *Neural Computation* 9 (1997): 1735-1780.
> >
> > Kingma, Diederik P. and Jimmy Ba. “Adam: A Method for Stochastic Optimization.” ICLR (2015).
> >
> > Angermueller, Christof, David Dohan, David Belanger, Ramya Deshpande, Kevin Murphy and Lucy J. Colwell. “Model-based reinforcement learning for biological sequence design.” *ICLR* (2020).

---

> ### Author Response · Authors · 2021-11-16
> **Hoping that our response could address your concern**
>
> We would appreciate it if you can let us know if our response has addressed your concern and thus improved your assessment of our work. We look forward to hearing from you!

---

> > ### Comment · Reviewer_JsdK · 2021-11-19
> > **Changing my review**
> >
> > I have increased my rating to 'weak accept'. I appreciate all of the extra experiments you did and clarifications on related work. While I appreciate that the paper establishes a coherent framework that allows many different optimization approaches, I still wish there was more of a focus on a single method that the authors think is best. That said, I think the paper is above the bar for acceptance and that the ICLR community would benefit from it.

---

> > > ### Author Response · Authors · 2021-11-20
> > > **Thank you very much for your feedback**
> > >
> > > Thank you very much for your feedback. We shall improve our work according to your suggestions.

---

### Official Review · Reviewer_LzqG · 2021-11-04

**Correctness:** 4
**Technical Novelty And Significance:** 4
**Empirical Novelty And Significance:** 2
**Recommendation:** 8
**Confidence:** 3

**Main Review:**

The link described in this work is interesting and the novel algorithms proposed contain significant differences to existing sequence optimization techniques. Empirical results support claims that these novel algorithms are interesting and bear consideration for future design efforts.

Strengths
- Both Iterative Scoring and Iterative Ratio lead heavily on supervised learning (outputting low-dimensional predictions) compared to many existing design algorithms. Training regression models instead of likelihood models on protein sequence space could yield useful empirical advances, making this an interesting contribution.
- The empirical evaluations are on well-known datasets and baselines appear to be used correctly. Results align with the conclusions of the paper.
- Drawing a distinction between "forward modeling" and "backward modeling" could provide generally useful language for the community and enable communication of ideas.

Weaknesses
The main weakness is in presentation of the link itself. In general the link seems to be "correct" in the sense that it is meaningful and consistently applied to link algorithms. However its exposition does give clear intuition for what is going on. I recognize this is not easy and try to provide some useful feedback below.
- On the one hand, the quantities (theta, x) have clear distinctions as parameters and data. On the other hand, the quantities (E, m) are a set of sequence and a sequence and do not have such a clear distinction. It is quite hard to see how the set E pops out of the mapping T described beneath Table 1.
- The notation p(E | m) = p(m \in E | m) is very confusing. The function p(E | m) reads like a probability distribution over subsets of sequence space, not a distribution over sequence space restricted to the subset E. I recommend finding a better notation here. This comment is based on equation 3, which I could be misunderstanding.


**Summary Of The Paper:**

The authors describe a mapping from likelihood free inference to black-box sequence optimization, then use this mapping to link common algorithms in both fields. They go on to describe novel black-box sequence design algorithms induced by known LFI algorithms. Empirical results show their methods are competitive on standard datasets.

**Summary Of The Review:**

The contribution of algorithms which heavily rely on regression / classification to guide sequence design is interesting, especially since they are the product of a general mechanism for producing sequence optimization methods. The empirical results seem sound. The exposition of the model could use work to help readers have a crisper sense of how probabilistic modeling gets linked to a problem setting where the only randomness is in experimental noise.

---

> ### Author Response · Authors · 2021-11-13
> **Response to Reviewer LzqG**
>
> Thank you for your insightful comments. Below we provide responses to the questions. We first point out that, as explained in the footnote of Page 2, we slightly overuse the notation for the event $\{m ∈ \mathcal{E}\}$ and the sequence set $\mathcal{E}$. This may not be a good usage, and we shall improve this in our final version.
>
> 1. It is true that $\theta$ and $x$ can be seen as parameters and data which are distinctive from each other and $\mathcal{E}$ is more closely related to $m$ in some sense. But here we provide another perspective. Notice that $x\sim p(x|\theta)$, meaning one can also see $x$ as a "result" or property of $\theta$. Similarly, the event $\{m ∈ \mathcal{E}\}$ can be seen as some property of sequence $m$ (for example, a desired specific chemical property).
> 2. Sorry about our misunderstanding notation. What we actually mean is $p(m\in\mathcal{E}|m)$ rather than a distribution of some sequence sets. We will consider using different notation for $\mathcal{E}$ and $\{m\in\mathcal{E}\}$ in our final version, as the reviewer suggests :)
>
> We are happy to answer any other questions you may have.

---

### Decision · Program_Chairs · 2022-01-20

**Decision:**

Accept (Spotlight)

**Comment:**

The paper investigates various approaches, and a unifying framework, for sequence design. There were a variety of opinions about the paper. It was felt, after discussion, that the paper would benefit from a sharper focus, and somewhat suffers from being overwhelmed by various approaches, lacking a clear narrative. But overall all reviewers had a positive sentiment, and the paper makes a nice contribution to the growing body of work on protein design.